# Local heating of radiation belt electrons to ultra-relativistic energies

Hayley J. Allison 1✉ & Yuri Y. Shprits[1,2,3]

Electrically charged particles are trapped by the Earth's magnetic field, forming the Van Allen radiation belts. Observations show that electrons in this region can have energies in excess of 7 MeV. However, whether electrons at these ultra-relativistic energies are locally accelerated, arise from betatron and Fermi acceleration due to transport across the magnetic field, or if a combination of both mechanisms is required, has remained an unanswered question in radiation belt physics. Here, we present a unique way of analyzing satellite observations which demonstrates that local acceleration is capable of heating electrons up to 7 MeV. By considering the evolution of phase space density peaks in magnetic coordinate space, we observe distinct signatures of local acceleration and the subsequent outward radial diffusion of ultra-relativistic electron populations. The results have important implications for understanding the origin of ultra-relativistic electrons in Earth's radiation belts, as well as in magnetized plasmas throughout the solar system.

[1] GFZ German Research Centre for Geosciences, 14473 Potsdam, Germany. [2] Institute of Physics and Astronomy, Universität Potsdam, 14469 Potsdam, Germany. [3] Department of Earth, Planetary, and Space Sciences, University of California, Los Angeles, CA 90095, USA. ✉email: haylis@gfz-potsdam.de

Discovered at the start of the space age, the Van Allen radiation belts consist of relativistic ($\gtrsim$500 keV) and ultra-relativistic ($\gtrsim$3MeV) electrons[1,2]. Energization mechanisms for these particles remain a topic of intense debate. Historically, inward radial diffusion was considered as the main acceleration process, whereby a source population at large radial distances diffuses toward the Earth due to electric and magnetic field fluctuations[3,4]. By moving into regions of stronger magnetic field, the kinetic energies of the particles are increased. However, this mechanism alone was shown to be insufficient to fully explain observed enhancements in the relativistic electron flux[5]. Later work demonstrated that electrons could also be accelerated locally via resonant interactions with naturally occurring whistler-mode electromagnetic waves[6], primarily very low frequency chorus waves[7], which can energize electrons to greater than a few MeV.

Distinguishing the relative importance of local acceleration and inward radial diffusion in energizing radiation belt electrons is possible by considering the radial profile of the electron differential flux divided by the particle momentum squared[8] (given in the magnetic coordinates that constrain electron motion: $\mu$, $K$, and $L^*$—see Supplementary Note 1), a quantity known as phase space density. The $L^*$ coordinate is the radius (in Earth radii) of a particle's drift around the Earth if the magnetic field adiabatically relaxed to a dipole configuration. Radial diffusion moves electrons across $L^*$ while conserving $\mu$ and $K$. The resulting radial profile is sloped, lacking growing peaks. Conversely, local acceleration occurs at constant $L^*$ whilst altering $\mu$ and $K$, producing growing peaks. By considering the evolution of phase space density across $L^*$, studies have mostly shown local acceleration to be the primary mechanism for generating 1–3 MeV electrons[8–10]. However, above energies of ~3 MeV, current theory indicates that acceleration via resonant interactions with chorus waves tends to decline[11,12] and how electrons come to be heated to higher energies is unclear.

Variable factors, such as the electron number density and the duration of chorus activity, influence the maximum energy achieved by local acceleration[5,12] and it is therefore possible that chorus interactions may be responsible for >3 MeV enhancements. Thorne et al.[6] used a 2-D diffusion model at fixed $L^* = 5$ and found that, under low plasma density conditions, chorus acceleration reproduced observed enhancements up to energies of ~7 MeV. However, to date, there have been no observations of local acceleration acting for these ultra-relativistic energies and analysis of phase space density profiles across a range of $\mu$ values is required to determine whether local acceleration can generate these populations.

A multistep acceleration process has recently been suggested[13–15], with particles first accelerated to relativistic energies locally by chorus waves and the newly formed population then further energized by inward radial diffusion[16]. However, radial profiles of phase space density relating to >3 MeV electrons have not been extensively studied and it remains unclear if relativistic electrons are accelerated to >1 MeV and then transported inwards, conserving the first and second invariants, or whether they can be locally accelerated directly to ultra-relativistic energies. If the multistep process is indeed the generation mechanism for ultra-relativistic electrons, then growing phase space density peaks would first be seen at large radial distances and subsequently broaden due to radial diffusion.

Here, we present observations from the Van Allen Probe satellites which show local acceleration acting to energize electrons to up to ultra-relativistic energies. For a broad range of the first adiabatic invariant, $\mu$, the evolution of electron phase space density across the $L^*$ parameter shows signatures of local acceleration. Radiation belt electrons with energies up to 7 MeV can therefore be enhanced in situ.

## Results

**Overview of the storm event.** An intense geomagnetic storm occurred on the 9 October 2012, during which the Relativistic Electron–Proton Telescope (REPT)[17] on the NASA Van Allen probes recorded flux enhancements across a range of energies, including at 7.7 MeV (Fig. 1). The storm was triggered by two periods of largely southward directed interplanetary magnetic field (Fig. 1h) and is characterized by two decreases in Dst, a measure of the geomagnetic storm intensity (Fig. 1i). During the second decrease in Dst, the electron flux showed prompt energization. As shown in Fig. 1b, early on the 9 October the 2.6 MeV flux at $L^* = 4$ rose by three orders of magnitude in less than 12 h, signifying an intense acceleration event. Several studies have previously focused on this storm, demonstrating that local acceleration mechanisms were both active during the period[6,18] and provided the primary energization process for $\mu = 3433$ MeV G$^{-1}$, $K = 0.11$ $R_E$G$^{1/2}$ electrons[9]. However, the electron energy corresponding to a particular $\mu$ changes with $L^*$ (Fig. 2) and analysis over a range of $\mu$ is required to determine how electrons at different energies come to be heated. In Fig. 1 enhancements in the 6.3 and 7.7 MeV flux are seen around $L^* = 4$ and, from Fig. 2, therefore correspond to $\mu$ values of 8000–12,000 MeV G$^{-1}$ for $K = 0.11$ $R_E$G$^{1/2}$. Here, we follow the work of previous authors and use $K = 0.11$ $R_E$G$^{1/2}$ which is sampled over a broad range of $L^*$ and generally corresponds to equatorial pitch angles greater than 45°[9,10,13] (see also Supplementary Fig. 13).

A second geomagnetic storm occurred on the 13 October, and crucially, in the 2 days following the storm, the flux at 6.3 and 7.7 MeV exhibited a second enhancement. The combination of the two successive geomagnetic storms resulted in increased levels of ultra-relativistic radiation belt electrons, which persisted for more than a week, presenting an interesting period for study.

**Analysis of phase space density.** In previous work, phase space density at fixed $\mu$ and $K$, measured through a number of outbound and inbound legs of the Van Allen Probes orbits, were analyzed for temporally growing peaks[9,10]. Generally, only a single $\mu$–$K$ pair is studied in this manner, owing both to the challenges involved in calculating phase space density, and because measurements of >6 MeV ultra-relativistic electrons are seldom distinguishable from the background level of the instrument. Focus has primarily been on lower values of $\mu$ (<6000 MeV G$^{-1}$), leaving the dynamics of high energy particles largely unexplored. Here, we study the October 2012 period, where 7.7 MeV electron flux values, greater than the background threshold [16], are observed during multiple days, and calculate the phase space density values relating to several different $\mu$, contrasting how electrons are enhanced over a broad range of energies.

Figure 3 is a schematic illustration showing the evolving $L^*$ coverage of phase space density values within a factor of five of the maximum phase space density, for various acceleration mechanisms. In this format, inward radial diffusion appears as a contour which gradually expands to smaller $L^*$, whilst also extending out to the last $L^*$ sampled (Fig. 3a). Local acceleration, on the other hand, presents as an enhancement appearing away from the maximum $L^*$, remaining at approximately the same location (Fig. 3b). However, peaks formed by local acceleration are likely to be redistributed by radial diffusion. A localized phase space density peak, formed in the heart of the outer radiation belt and gradually diffusing to higher $L^*$ is shown in Fig. 3c, while a peak at larger $L^*$, radially diffusing inwards, is shown in Fig. 3d.

The temporal evolution of the $L^*$ extent of phase space density maxima during the October 2012 period is considered for eight values of $\mu$ (Fig. 4), in the format of Fig. 3, highlighting

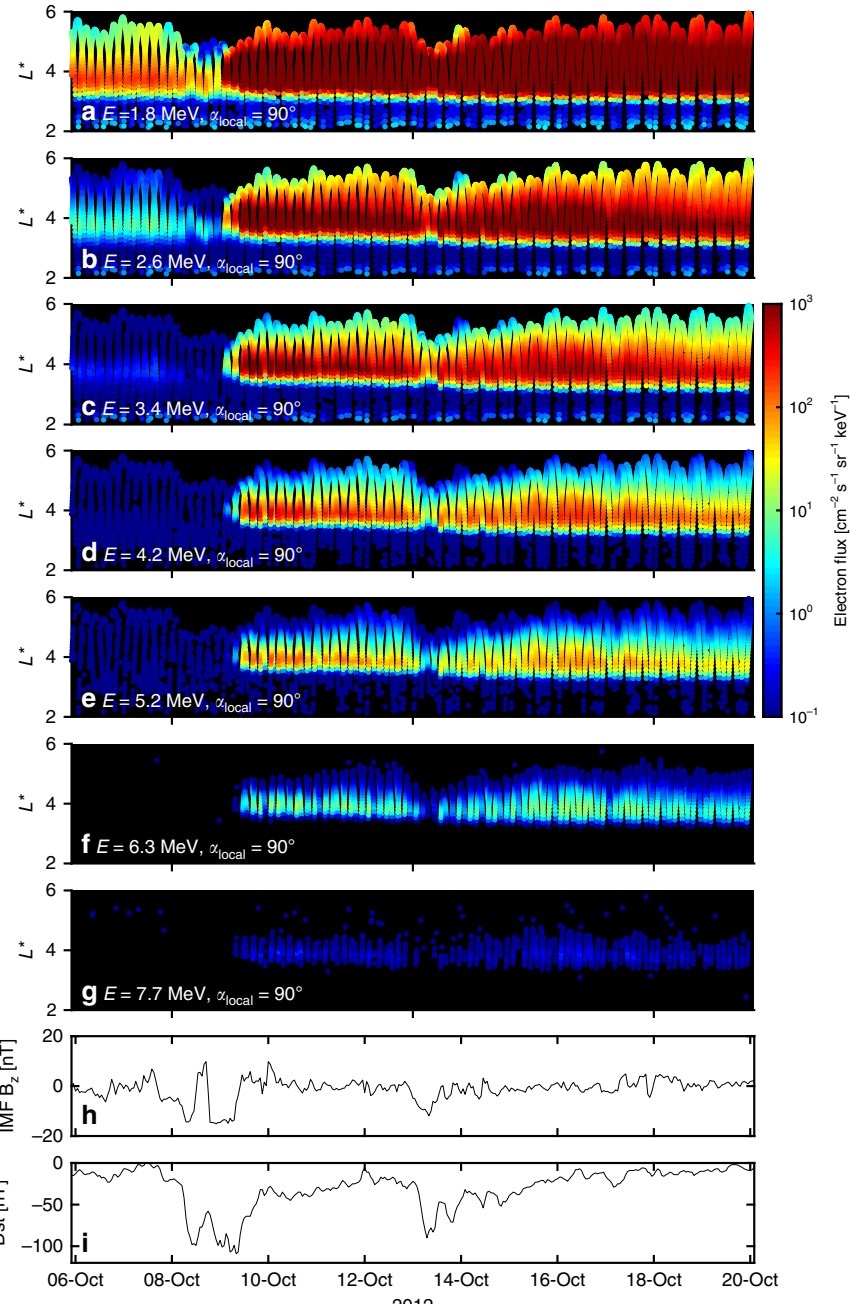

**Fig. 1 An overview of the radiation belt electron flux across a number of energies during 6–20 October 2012. a–g** Show the electron flux measured by the REPT instrument on NASA's Van Allen Probe A and B for electron energies from 1.8 to 7.7 MeV. The flux corresponds to a local pitch angle of 90° and the associated $L^*$ coordinate has been calculated using the TS07 magnetic field model[27]. Only electron flux measurements above the noise floors[28] of REPT[17] have been shown. **h** Shows the evolution of the north–south component (relative to the Earth's magnetic field) of the interplanetary magnetic field (IMF $B_z$) for the period, and **i** shows the disturbance storm time index (Dst), a measure of the geomagnetic storm intensity. The geomagnetic activity on the 9 October, resulting from a period of strongly negative IMF $B_z$, produced an enhancement in the electron flux over a broad energy range, including at 7.7 MeV.

enhancements. The multistep formation method for >3 MeV electrons, suggested by several authors[16,19], would appear as the profile shown in Fig. 3d. However, over a time-span of several days, the profiles for both storms shown in Fig. 4a more closely resemble local acceleration followed by outward radial diffusion (Fig. 3c) during the recovery period. For the enhancement during the first storm, contours are concentric with increasing $\mu$ and do not extend to the last $L^*$ sampled (black region) indicating that a peak arose for a broad range of energies at the same location. A temporal shift is observed, with the peak forming at high energies (up to 7 MeV)

later than at lower energies, consistent with energy diffusion at this point in space. Note that for $\mu = 10,000$ MeV G$^{-1}$, the $L^*$ extent of the phase space density enhancement rapidly shifts inwards across $\sim 0.3 L^*$. This $L^*$ variation may be the result of inward radial diffusion or may indicate local acceleration from a shifting source region. From Fig. 2, an inward radial motion from $L^* \sim 4.5$–4.2 constitutes an energy increase of $\sim 600$ keV at $\mu = 10,000$ MeV G$^{-1}$, and so a source population of $\sim 6.4$ MeV would be needed to produce the 7 MeV enhancement at $L^* = 4.2$. Van Allen Probes passes during this acceleration period (first region shaded gray) are

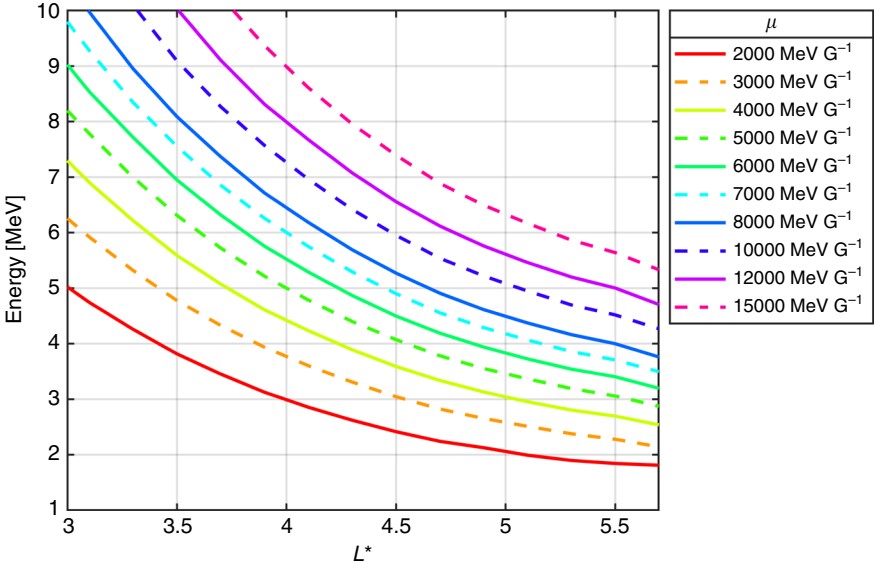

**Fig. 2 The average electron energy, across $L^*$, for various values of $\mu$.** Average energies corresponding to $\mu$ values between 2000 and 15,000 MeV G$^{-1}$, for the 6–20 October 2012 period are shown. For all values of $\mu$, $K = 0.11\ R_E G^{1/2}$. The $\mu$ value has been calculated using the locally measured magnetic field, while $K$ and $L^*$ were calculated using TS07[27].

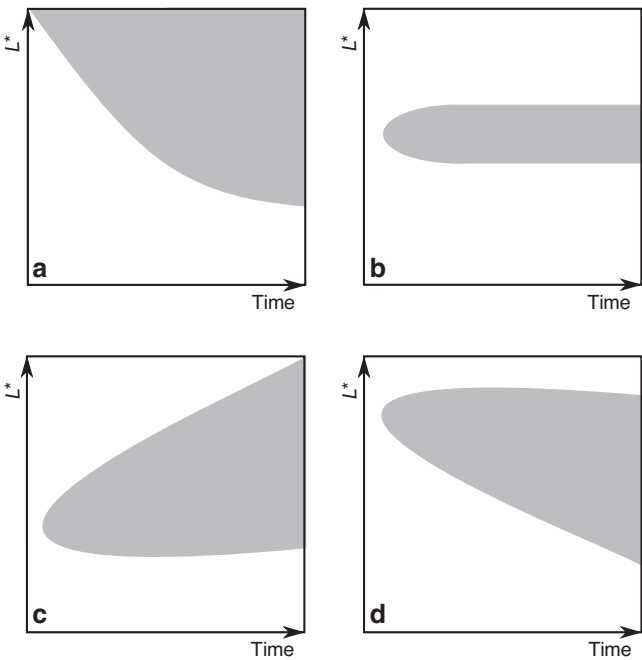

**Fig. 3 An illustration of phase space density enhancements in $L^*$ and time for different acceleration mechanisms.** The gray contour marks the evolving $L^*$ extent of the phase space density enhancement. Profiles for inward radial diffusion and local acceleration are shown in (**a**) and (**b**), respectively. The combination of local acceleration and outward radial diffusion is shown in (**c**) and local acceleration coupled with inward radial diffusion in (**d**).

analyzed more closely for $\mu = 8000$ and 10,000 MeV G$^{-1}$ electrons in Fig. 5a, b.

## Discussion
The phase space density along the inbound pass of Van Allen Probe A and outbound pass of probe B are shown prior to the enhancement (black lines, Fig. 5a, labeled 20:45 to 23:10 8

October and 22:35 8 October to 01:30 9 October). These phase space density values are close to the background thresholds of the instrument but are included here to illustrate the upper bound of the pre-existing phase space density level. Over the next few passes of both probes, a peak emerges which grows with time. The broadening of the phase space density peak is likely a result of both inward and outward radial diffusion, or a dynamically evolving acceleration region, however, the presence of a growing peak is a signature of local acceleration. The primary $L^*$ coverage of the peak is $L^* = 3.8$–5.0, corresponding to electrons up to ~7.5 MeV.

As was also shown in Fig. 4a, at $\mu = 10,000$ MeV G$^{-1}$, enhancements were observed at later times than at $\mu = 8000$ MeV G$^{-1}$. Reeves et al.[9] reported that the $\mu = 3433$ MeV G$^{-1}$ phase space density showed an enhancement between the 8 and 9 October (we also see similar behavior at this $\mu$. See Supplementary Fig. 3a), while in Fig. 5, increases at $\mu = 8000$ and 10,000 MeV G$^{-1}$ were primarily seen at later times. The enhancement therefore occurred first at lower $\mu$ values. During local acceleration, electrons are accelerated across energy space, and an observed delay for enhancements to reach higher values of $\mu$ is consistent with this heating mechanism.

During the 07:20 to 10:35 9 October outbound pass of Van Allen Probe B, the phase space density peak had increased at $\mu = 8000$ MeV G$^{-1}$ and an enhancement was also seen for $\mu = 10,000$ MeV G$^{-1}$. Negative phase space density gradients were observed around $L^* \sim 4.7$ for both values of $\mu$, however, there is a very sharp increase toward the end of the orbit, outside $L^* \approx 4.8$ (see Supplementary Fig. 12). Supplementary Note 7 examines this pass in more detail and it was concluded that the enhanced phase space density at $L^* \gtrsim 4.8$ originates from an observed increase in the $B_z$ component of the magnetic field, that is not fully captured by the field model, together with an inversion of the local pitch angle distribution. Note that the increase is preserved at a similar $L^*$ on the following inbound pass of probe B and is not observed at all by probe A. As it would appear that the enhancement at $L^* \approx 4.8$ is an artifact of field model, we do not show this in Fig. 5a, b. In addition, Supplementary Fig. 9 shows that the last closed drift shell was at $L^* = 5.7$ for $K = 0.11\ R_E G^{1/2}$. The lowest energies at $\mu = 10,000$ MeV G$^{-1}$ and $\mu = 8000$ MeV G$^{-1}$ are

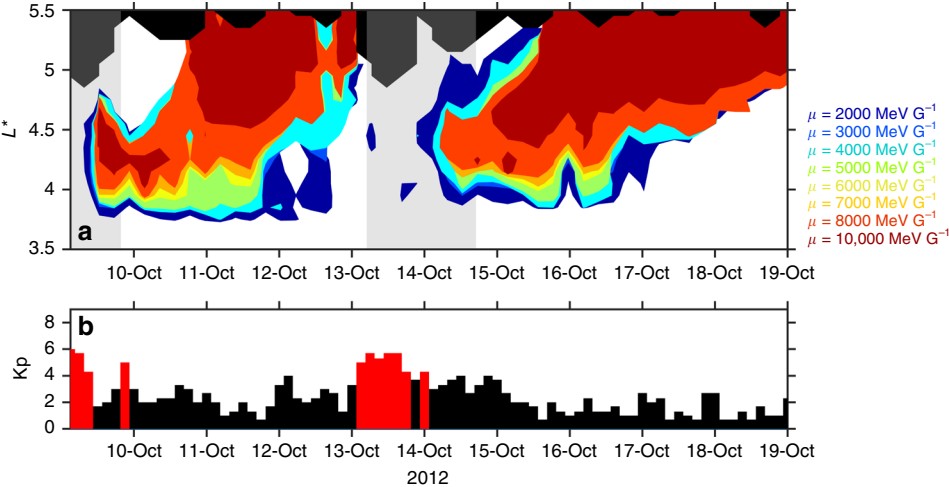

**Fig. 4 Evolution of the L\*-coverage of phase space density enhancements during October 2012.** In (**a**), contours show the $L^*$ extent of phase space density values that are within half an order of magnitude of the maximum phase space density for the period (at a specific value of $\mu$ ranging from 2000 to 10,000 MeV G$^{-1}$). To form the contours, the phase space density values (Supplementary Fig. 2) were binned into 5-h intervals and by 0.1 $L^*$. By layering these contours for different values of $\mu$, acceleration regions over a broad energy range are revealed. $L^*$ values not sampled by the Van Allen Probes are shown in black. The phase space density profile shows a peaked profile between $L^* \approx$ 4.0–4.5, at which $\mu = 8000$ and 10,000 MeV G$^{-1}$ correspond to ~5.4–6.5 and ~6.0–7.3 MeV, respectively (see Fig. 2). The $K_p$ index, shown in (**b**), provides a measure of geomagnetic activity and is derived from ground magnetometer measurements. $K_p < 4$ is shown as black bars and $K_p \geq 4$ as red bars to highlight periods of geomagnetic disturbance.

therefore ~4.2 and ~3.8 MeV, respectively (from Fig. 2). If the rapid change in phase space density does not originate from magnetic field discrepancies, then this raises an interesting question on the origin of the population, but is beyond the scope of this study.

After the phase space density enhancement, the peak broadens over time and becomes more of a plateau (see Supplementary Fig. 4). The radiation belt electrons therefore experienced radial diffusion, transporting them outwards. While the maximum phase space density contours shown in Fig. 4 expand toward higher $L^*$ for all values of $\mu$ shown, after 11 October, for $\mu \leq 5000$ MeV G$^{-1}$, broader $L^*$ profiles are observed than for higher $\mu$. The different behavior exhibited by $\mu \leq 5000$ MeV G$^{-1}$ and $\mu > 5000$ MeV G$^{-1}$ electrons may be indicative of energy dependent loss mechanisms, such as electromagnetic ion cyclotron waves[20,21].

Following the second storm in October 2012, the phase space density again shows peaked structures centered on similar $L^*$ for a broad range of $\mu$ (Fig. 4a). As for the first storm, we further analyze this period by considering individual passes of the Van Allen probes (Fig. 5c and d). Prior to the acceleration, magnetopause shadowing[22] and outward radial diffusion[23] greatly reduced the pre-existing levels of phase space density and the initial profile is shown in black. At later times, a peak forms for both $\mu = 8000$ MeV G$^{-1}$ and $\mu = 10,000$ MeV G$^{-1}$, centered at $L^* \sim 4$. Over the subsequent passes, this initial peak broadens in $L^*$ before growing further, concentrated around $L^* \approx 4.3$. While the broadening suggests a contribution from radial diffusion, the growing peaked structure indicates that local acceleration was an active process during the second storm, forming the enhancements seen at $\mu = 8000$ MeV G$^{-1}$ and $\mu = 10,000$ MeV G$^{-1}$. After the localized increase in phase space density, a radial shift toward larger $L^*$ can be seen (Fig. 4) starting on 15 October. The rate of outward radial motion appears relatively invariant with $\mu$, suggesting energy independent diffusion, consistent with radial diffusion driven by magnetic pulses[3] rather than electrostatic pulses, or an azimuthal electron field power spectral density that is independent of frequency[24]. After 17 October, the dynamics of the peaks exhibit more variation with $\mu$.

Phase space density peaks may, in some cases, also arise due to a rapid enhancement at the outer boundary followed by fast loss[25]. The twin Van Allen probes sample only two locations at any one time and, owing to the spatial and temporal limitations, it is possible that a rapid enhancement and loss could occur that is missed by the spacecraft sampling. As discussed in Supplementary Note 5, the orbits of the two Van Allen Probes impose times restrictions on this process, making such a scenario unlikely. Furthermore, the last closed drift shell location is at $L^* \sim 5.5$–6.5 for the first storm and $L^* \sim 6$–6.5 for the second (Supplementary Note 6). The trends shown in Fig. 2 then suggest that the minimum energy of the source population required for the observed 7 MeV enhancements is likely to be several MeV. Observations from geostationary orbit of the >2 MeV flux measured by the Geostationary Observational Earth Satellites (GOES) 13, 14, and 15 are shown in Supplementary Figs. 10 and 11. When considered alongside the changes in the $L^*$ measured, these additional observations do not suggest a rapidly appearing source population for radial diffusion during either event. Furthermore, the GOES flux is higher in the days following the acceleration occurring at $L^* \sim 4$ than it was during the enhancement period, consistent with a locally accelerated source that diffuses outwards. However, we note that these are again in situ measurements and therefore also subject to certain spatial limitations. The GOES spacecraft may not have been appropriately situated to measure a source population during the period. In addition, because radiation belt electrons often show steep falling energy spectra, the integral energy channels may not necessarily show the source populations of >7 MeV electrons in the outer belt.

Figures 4 and 5 effectively demonstrate that local acceleration is capable of heating electrons to ~7 MeV as the phase space density profiles show signatures of local acceleration during both of the geomagnetic storms considered. The phase space density enhancements for higher energies followed the enhancements at lower energies. In Supplementary Note 8, additional analysis establishes that locally growing peaks are also observed for lower values of $K$, corresponding to radiation belt electrons confined closer to the equator. Furthermore, as the values of $K$ and $L^*$ are

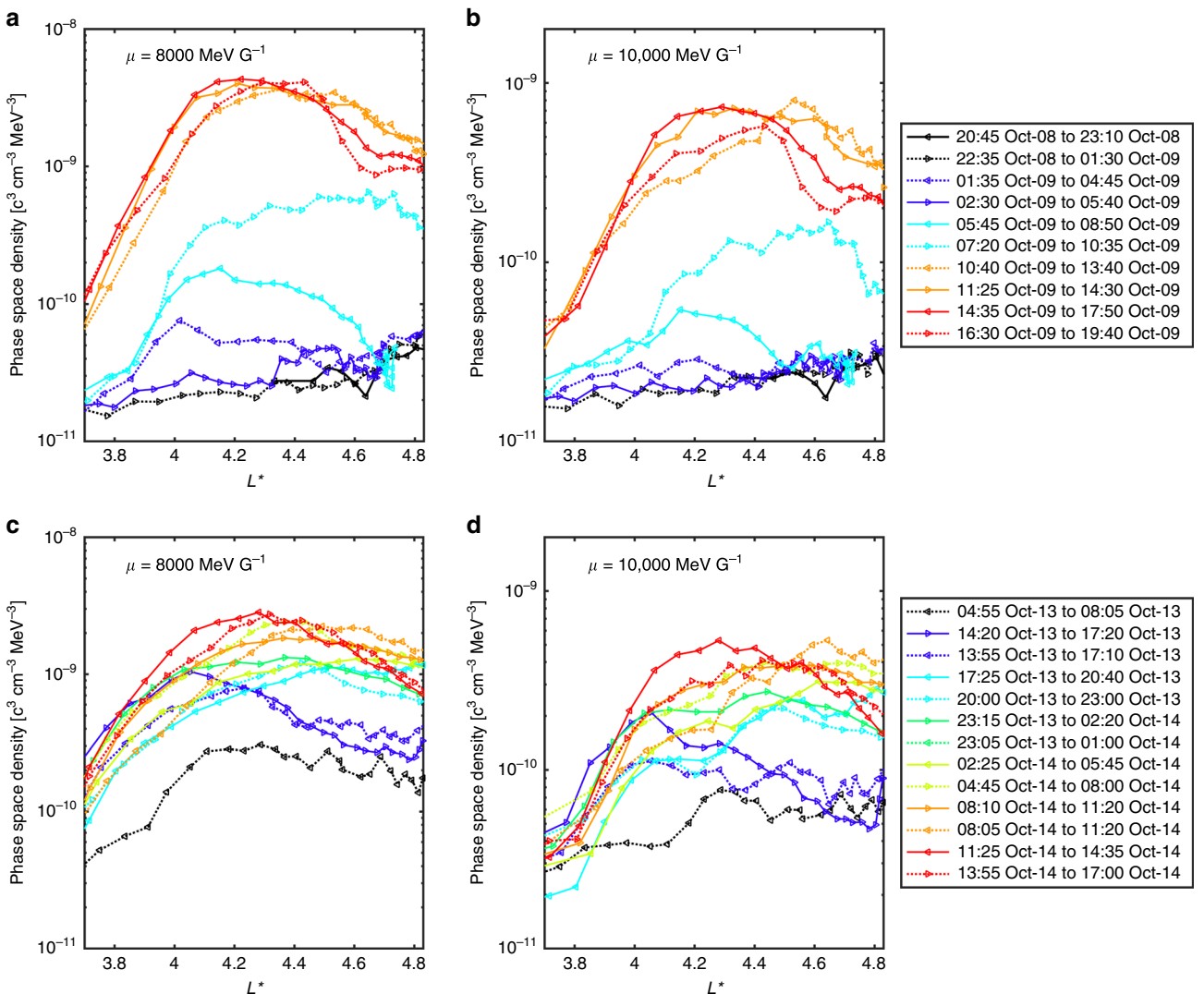

**Fig. 5 Radial profiles of phase space density during the first and second storm.** Phase space density—$L^*$ profiles measured by Van Allen Probe A (solid lines) and B (dotted lines) are shown. **a**, **b** Show the phase space density during the first enhancement period for $\mu = 8000$ MeV and $\mu = 10{,}000$ MeV, respectively. **c**, **d** Again show the phase space density for $\mu = 8000$ MeV and $\mu = 10{,}000$ MeV, respectively, but for the second enhancement period. Growing peaks are observed, indicative of local acceleration. All phase space density values correspond to $K = 0.11\ R_E G^{1/2}$. Arrows mark the direction of spacecraft travel and we have used the maximum radius of the orbit (from the $x$, $y$, $z$ coordinates) to distinguish between the inbound and outbound sectors of the pass. For each pass, the time of the first and last measurement shown is used to label the phase space density—$L^*$ profile. Not all passes are shown here and for additional passes see Supplementary Figs. 5 and 6.

dependent on the magnetic field model chosen, results using an additional two field models are also presented (see Supplementary Note 9) and, once again, growing peaks are observed in the radial phase space density profile. Our results demonstrate that local acceleration had a significant effect on radiation belt particles during both of the storms in October 2012, acting on electrons up to 7 MeV. In the radiation belt region, local acceleration introduces radial gradients in phase space density and so is always accompanied by both outwards and inwards radial diffusion. Locally heating electrons to ~7 MeV provides a very high energy "source population" for inwards radial diffusion and could therefore help explain the occurrence of ~10 MeV electrons in April–May 2017[16].

A recent study by Zhao et al.[15], considered the acceleration of ultra-relativistic electrons via a statistical analysis of events during the Van Allen Probe era. The results were consistent with a two-step acceleration process, where locally heated electrons at large $L^*$, beyond the Van Allen Probes apogee, are radially diffused

inwards to reach energies of 7 MeV in the outer radiation belt. While the combination of local acceleration and radial diffusion produces 7 MeV enhancements[15], the Van Allen Probe observations for the two storms shown in this study demonstrate that local acceleration can also act directly up to 7 MeV energies. The local energization mechanism responsible for generating 7 MeV electrons in the heart of the outer radiation belt, be that acceleration by chorus waves or some other process, presents an interesting focus for future research. Longer term analysis and statistical studies can be used to better understand the conditions leading to acceleration. Datasets formed via data-assimilation techniques may be useful for this purpose. Long term observations of the ultra-relativistic component of Earth's radiation belts demonstrate that ≥7 MeV electrons are a relatively rare phenomenon, occurring far less frequently than enhancements at 1 or 2 MeV[1]. It therefore follows that the circumstances leading to multi-MeV enhancements could be unusual, requiring specific conditions. Our results highlight that wave-particle interactions

can provide the primary acceleration mechanism for electrons up to ultra-relativistic energies, a finding applicable to magnetized plasmas throughout the solar system.

## Methods

**Calculating phase space density**. The Van Allen Probes REPT data[17] provide electron flux in 12 energy channels, ranging from 1.8 to 20 MeV, with a pitch angle resolution that is achieved by spin sectoring. In this study we used 5 min averaged REPT data to calculate the phase space density. Flux measurements from REPT were converted to phase space density by dividing by the relativistic particle momentum squared. The first adiabatic invariant, $\mu$, was calculated for each energy and pitch angle channel of the REPT instrument using local magnetic field observations from the Electric and Magnetic Field Instrument Suite and Integrated Science also on-board the satellites[26]. The local magnetic field observations were also processed into 5 min intervals, from which the median measurement was taken. $K$ and $L^*$ values were then computed for each local pitch angle and satellite position using the TS07[27] magnetic field model with the publicly available International Radiation Belt Environment Modeling library (https://craterre.onera.fr/prbem/irbem/description.html).

To determine the phase space density at the target $\mu$ and $K$ values, we first interpolate the logarithm of the phase space density, and the corresponding $\mu$ and $L^*$ values, to the target value of $K$. The logarithm of the phase space density is then interpolated again to the chosen value of $\mu$. We never use phase space density data that do not directly surround the chosen values of $\mu$ and $K$. If one or both of the immediate neighbors to the target invariant values has no corresponding phase space density data, then the interpolated phase space density was returned as missing data. A model pitch angle distribution was not used when interpolating to the target $K$ value. However, as shown in Supplementary Fig. 1, using a $f(\alpha) = A\sin(\alpha) + C$ function between the two pitch angle channels immediately surrounding the target $K$ has a very limited effect on the results.

## Data availability

The Van Allen Probes data presented in this study are publicly available and can be downloaded from https://rbsp-ect.newmexicoconsortium.org/data_pub/. In addition, the Van Allen Probes data can be visualized with the Van Allen Probes Science Gateway at https://rbspgway.jhuapl.edu/home_overview. Dst, $B_z$, and $K_p$ values are from the NASA OMNIWeb data explorer, accessible at https://omniweb.gsfc.nasa.gov/form/dx1.html.

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

## Acknowledgements

This work was funded by Helmholtz-Gemeinschaft (HGF) 10.13039/5 011 00001666 and Deutsche Forschungsgemeinschaft (DFG) through Grant CRC 1294 "Data Assimilation" Project B06. Y.Y.S was funded by NASA grant NNX15AI94G. We acknowledge Nikita Aseev for their useful discussions regarding the work.

## Author contributions

H.J.A wrote the paper, performed the analysis, and constructed the figures for the paper. Y.Y.S conceived the idea for the study, suggested the format for Fig. 4, provided extensive analysis support, and edited the paper.

## Funding

## Competing interests

The authors declare no competing interests.
