## [Peer Review File · Nature Communications]

Reviewers' comments:

Reviewer #1 (Remarks to the Author):

Review of Local Heating of Radiation Belt Electrons to Ultra-relativistic Energies by Allison & Shprits

The major claim of the paper is that the October 2012 set of storms represented a period wherein radiation belt electrons were accelerated up to ultrarelativistic energies by local acceleration alone. The topic is of prime interest, since disentangling acceleration processes is crucial to understanding the overall dynamics of the system, and tying those dynamics to features in the solar wind that ultimately drive this variability. The conclusions would be of great interest to the community; unfortunately, the data does not support the conclusions made here. I have great concern around the interpretation of the phase space density data derived that serve as the major focus of this paper. I have listed major issues (as well as more minor), but overall do not see how the main conclusion is arrived at from the data presented in this paper.

I also have concerns that the PSD data used in this paper is a derived quantity and is not available to the public, thus restricting any reproducibility of results. The Acknowledgments claim that all data used is available publicly, but this refers to the spin-sectored flux data, not PSD. PSD was calculated using (I assume) a model energy spectrum and pitch angle distribution, but that is not specified in the paper or supplemental, so there is no way for any person to follow the methodology and reproduce these results.

Major issues:

Line 73-75: "Here, we study the October 2012 period, where the flux of >7 MeV electrons remained above the background threshold [16] for more than seven days..." I don't think the 7.7 MeV flux was above background continuously over the time period of these two storms. Can you show evidence of this? The 6.3 MeV channel very much looks like it was above background, but 7.7 MeV looks count-starved during some of this time.

Line 88-89: "However, for both storms, Fig 3 (top) more closely resembles local acceleration followed by outwards radial diffusion (Fig 2c)." ☒ Not true. During the *recovery period* of the first storm, we can see outward radial diffusion taking place. But during the main phase of the storm, or

what you shade gray as the acceleration region, we see evidence of inward radial diffusion (Oct 9 – 10.5).

Line 99-100: “Reeves et al. [12] reported that the $\mu = 3,433$ MeV/G phase space density showed an enhancement between the 8 and 9 October.” You have this data as well – do you also see the enhancement at ~ 3000 MeV/G?

Line 109: “After the first phase space density enhancement, the peak moved to larger L^* .” What does this mean? What first enhancement? If you look carefully at Fig 3b-c, you can see that the peak of PSD moves inward during the time shown. You don’t show the radial profiles after early on 10 Oct, so it’s difficult to see what happens after that but I suspect the peak broadens instead of really strongly peaking at higher L^* , such that the one “peak” can be picked out at higher L^* but really it’s more of a plateau. Need to see more information – please put in the paper or Supplemental.

Line 111-112: “While the maximum phase space density region expands towards higher L^* for all values of μ shown, for $\mu < 6,000$ MeV/G, an enhanced population remains around $L^* = 4$, not observed for higher values of μ .” What exactly are you referring to here? I don’t see this feature in any plots. In supplemental figure S7, I see enhanced PSD at $L^*=4$ for $\mu = 5000$ MeV/G, 9000 MeV/G *and* 12000 MeV/G. What is this difference you are talking about? Please clarify.

Line 118-122: Again, as for the first double-dip storm, the outward radial diffusion looks like it doesn’t play a role until after the main phase, during recovery (here, starting on Oct 15th). In Fig 3d-e, you can see the local peak growing from red to green traces. Then, from green through cyan to the first blue, the peak clearly shifts inward in L^* , indicating *inward* radial diffusion. This is during the acceleration period. After that, you get some complicated dynamics where the peak falls to lower PSD values, shifts outward, then rises and shifts inward again (final pink to red traces). This figure does indeed show the signature of inward radial diffusion, contrary to the final conclusions of the paper.

Line 124-126: Absolutely not. I see evidence of inward radial diffusion throughout both storms from the PSD radial profiles. How else do you explain the *inward* motion of PSD peaks?

Line 130-132: Part of this sentence holds: “Our results demonstrate that local acceleration had a significant effect on radiation belt particles during both of the storms in October 2012” The data shows this. However, the data does not support the last part of that sentence: “[local acceleration is] acting on electrons up to 7 MeV.” There is just no way to disentangle the effects of inward radial diffusion and local acceleration in this set of storms, since you see evidence of both processes happening. It might be possible to disentangle in other storm events.

Line 136-137: “occurring only a handful of times during the Van Allen Probes mission.” Not true – it has happened at least a couple dozen times, depending on your threshold for acceleration events. I suggest changing this wording.

Minor issues:

Line 35-36: Awkward language. “keeps electron’s L^* values” should be something like “electron’s drift within constant L^* ” or similar.

Line 36: Not sure why “producing” is emphasized; shouldn’t it be “growing” that is the crucial term here?

Line 54: More awkward language. “Consider Fig 1b” should be something like “As shown in Fig 1b”.

Line 77-79: I don’t understand. Fig 2 is a cartoon schematic, right? Why is it described as using “PSD values within half-an-order of magnitude of the maximum for the period”? Which period – the one studied here? I thought this was just a schematic representation – not using actual data. Maybe you meant to say “Fig 3”? In any case, I think Fig 2 should be described first before moving on to describe Fig 3.

Line 89: “outward radial diffusion”

Figures:

- Figure 1 needs to be completely redone – it is poor quality and hard to read labels.
- Figure 3 needs to be redone. The L^* labels need to be bigger in panel a. Resolution needs to be better. (This is true with all figures and may be an artifact of using PDF in LaTeX. Convert to .jpg or .png first and it should help.)

Reviewer #2 (Remarks to the Author):

This study, using the Van Allen Probes’ measurements of multi-MeV electrons, presents a unique way of analyzing electron phase space densities with a wide range of μ values. The results show that during an intense storm of October 2012, electrons are heated up to ~ 7 MeV by local acceleration. Thus, the authors suggest that the acceleration of ultrarelativistic electrons with energies up to ~ 7

MeV can be caused by local heating alone and a two-step process of local acceleration combining with inward radial diffusion, which is suggested by some previous studies, is not required in accelerating electrons to ultrarelativistic energies.

This topic on the acceleration of ultrarelativistic electrons is under-explored due to the lack of high quality data prior to the Van Allen Probes era. Fully understanding the acceleration of ultrarelativistic electrons will contribute to a more comprehensive understanding of the radiation belt dynamics and thus the magnetospheric physics. It will also shed light on the energy-dependent dynamics of radiation belt particles and will be of great interest to others in this field. This manuscript is concise and well-written. However, I still have some concerns about this study's novelty and methods.

First, as the authors also pointed out in their manuscript, several studies on the acceleration of ultrarelativistic electrons have been carried out in the Van Allen Probes era, including both data analysis and modeling. The storm this study focused on is also a widely-studied one. Specifically, Thorne et al. (2013), using a 2-D diffusion model with chorus wave heating, well reproduced the flux enhancements of electrons with energies up to ~ 7 MeV in the outer radiation belt during this Oct 2012 storm and demonstrated the dominant role of local acceleration in ultrarelativistic electron flux enhancements. This study by Allison et al. shows intriguing observational results confirming these results, however, the novelty may be somewhat compromised due to the previous publications.

Second, this study focuses on the acceleration of ultrarelativistic electrons up to ~ 7 MeV, which has been also emphasized by the authors in several places in the manuscript (abstract, line 132, etc.). However, the main results regarding the evolution of electron phase space densities focus on $\mu=8000$ MeV/G and $\mu=10000$ MeV/G, $K=0.11$ G1/2Re electrons. It was stated in the manuscript that at these μ and K values the energy of electrons correspond to ~ 6 MeV and ~ 7 MeV at $L^*\sim 4.2$ respectively. However, reading from their Figure S1, these μ values should correspond to ~ 5.5 MeV and ~ 6 MeV respectively, and ~ 7 MeV electrons should correspond to much higher μ values at $L^*\sim 4 - 4.5$ ($>\sim 12000$ MeV/G). It is possible that during some portion of this storm the magnetic field was more distorted and these μ values corresponded to higher energies at $L^*\sim 4.2$; however, the mismatch between texts and Figure S1 is still a concern and may confuse readers.

Third, the way of showing the spatiotemporal evolution of phase space density maxima for a range of μ values is innovative. However, it still suffers from the similar problem with the traditional way of looking at phase space density radial profile at fixed μ and K : without sufficient spatial coverage and time resolution, the results can be somewhat ambiguous. For example, a local peak could be caused by the local acceleration alone but could also be caused by inward radial diffusion combined with fast loss at high L^* . The authors should at least discuss more about this aspect.

Some other comments/questions:

1. The author pointed out the study by Katsavrias et al. (2019) on the importance of two-step acceleration on the ultrarelativistic electron flux enhancements in the April 2017 storm. However, an early study by Zhao et al. (2018) studied the same storm using both phase space density data analysis and radial diffusion modeling, showed the dominant role of inward radial diffusion in the acceleration of ~ 7 MeV electrons in the center of outer belt, and proposed a two-step acceleration mechanism for ultrarelativistic electron flux enhancements. Since this study by Zhao et al. (2018) focused on $K=0.1$ G1/2Re electrons instead of near-equatorially mirroring electrons with $K \leq 0.03$ G1/2Re in Katsavrias et al. (2019), it is more relevant to this study and should be included as a reference.

Zhao, H., Baker, D. N., Li, X., Jaynes, A. N., & Kanekal, S. G. (2018). The acceleration of ultrarelativistic electrons during a small to moderate storm of 21 April 2017. *Geophysical Research Letters*, 45, 5818–5825. <https://doi.org/10.1029/2018GL078582>.

2. Line 39 "... above energies of ~ 3 MeV, chorus wave acceleration declines...": since this study mainly focuses on the importance of chorus wave acceleration on ultrarelativistic electron flux enhancements, it would be helpful to have some discussions regarding the reason of chorus wave acceleration declining for ultrarelativistic electrons and why this does not affect the main result of this study.

3. As the results of this study heavily rely on the accuracy of phase space density calculation, the method used to calculate phase space densities and phase space coordinates will need to be provided in more detail. For example, how was the interpolation done and what assumptions have been used when interpolate the energy spectrum/pitch angle distribution? These information will be critical for readers to reproduce their results.

4. Figure S8 and S9: It would be better to show the corresponding μ values, 8000 MeV/G and 10000 MeV/G, as in the main manuscript, in order to make a more direct comparison.

Reviewer #3 (Remarks to the Author):

This paper examines radial phase space density profiles from two events in October 2012, examining what acceleration mechanism is responsible for the enhancement of ultra-relativistic electrons during this period. They conclude that for these events, local acceleration is responsible for heating these electrons up to 7 MeV.

Overall, the analysis and visualizations in this paper are appropriate and well done. In addition, the results could represent an important and influential contribution to the community. However, some additional analysis/discussion is needed to support the overall conclusions as currently written.

Major Comments:

At present, the paper suggests that the presented observations “unequivocally demonstrates that electrons are heated up to 7 MeV from local acceleration alone”. While the paper clearly demonstrates that local acceleration is active during these events and is likely the dominant mechanism, some additional discussion/analysis is needed to justify that local acceleration is solely responsible for the observed enhancements.

For the 9 October event, the authors state: “Throughout the enhancement, negative gradients were observed at $L^* > 4.5$, indicative of local acceleration.” However, looking at Figure 3b-c, there is a clear enhancement near apogee for the 9-Oct 8:45 and 12:40 passes. This coincides with the large enhancement at $L=4.25$. On a related note, I’m unclear why these enhancements near apogee don’t appear in Figure 3a.

Similarly, for the 14-15 October event: The 14-Oct 22:14:19 pass in purple for 10000 MeV/G shows a clear secondary peak at $L^* = 5.1$. This precedes some of the growing peaks at lower L^* observed in the subsequent passes (00:07 and 03:25).

Without observations at larger L^* , this suggests that the observed enhancements at low L^* could be due (at least in part) to rapid inward transport coupled with an on-off source process at higher L^* . This possibility and any constraints the observations would place on it should be discussed.

Minor Comments:

- Figure 2: This figure is a very nice visualization of the different acceleration mechanisms. My only minor suggestion would be to consider having the local acceleration peak appear at the same L^* in each panel. Relative to panel 1b, panel 1d appears at larger L^* and panel 1c at smaller L^* . If this makes the plots more difficult to read, this suggestion can be ignored.

- Figure 3: Marking these intervals in some way to show inbound vs. outbound passes would be useful.

Response to reviewers

Here we list each comment made by the reviewers in turn and respond in blue font below. All line numbers given correspond to the updated manuscript (track changes version).

We have supplied a zip folder containing processed .mat files of the Van Allen Probe phase space density data, adiabatic invariants, and flux values for the three magnetic field models used in the study (TS07, T89, and TS04). File names contain the field model name and the final letter of the file name (either 'a' or 'b') denotes which from which probe the data originated. Additionally, we have also included in the zip folder our Matlab function for interpolating to the chosen value of μ and K .

Reviewer #1 (Remarks to the Author):

Review of Local Heating of Radiation Belt Electrons to Ultra-relativistic Energies by Allison & Shprits

The major claim of the paper is that the October 2012 set of storms represented a period wherein radiation belt electrons were accelerated up to ultrarelativistic energies by local acceleration alone. The topic is of prime interest, since disentangling acceleration processes is crucial to understanding the overall dynamics of the system, and tying those dynamics to features in the solar wind that ultimately drive this variability. The conclusions would be of great interest to the community; unfortunately, the data does not support the conclusions made here. I have great concern around the interpretation of the phase space density data derived that serve as the major focus of this paper. I have listed major issues (as well as more minor), but overall do not see how the main conclusion is arrived at from the data presented in this paper.

I also have concerns that the PSD data used in this paper is a derived quantity and is not available to the public, thus restricting any reproducibility of results. The Acknowledgments claim that all data used is available publicly, but this refers to the spin-sectored flux data, not PSD. PSD was calculated using (I assume) a model energy spectrum and pitch angle distribution, but that is not specified in the paper or supplemental, so there is no way for any person to follow the methodology and reproduce these results.

Firstly, we would like to thank the reviewer for his/her careful review of the manuscript. We thoroughly appreciate the time taken to evaluate and check the work and results as well as the comments provided.

We understand that they have significant concerns regarding the main conclusion of the work. We stress that the main conclusion of the paper is not that radial diffusion plays no part during the two storms but rather that signatures of local acceleration are observed for electrons with energies up to 7 MeV, demonstrating that local acceleration can directly act at these energies. We have tried to address all of their concerns, and feel that, as a result, the paper and findings have been significantly improved. We are pleased that the reviewer thinks that the topic of the paper is of prime interest and we hope that with the changes made to the paper, they may agree with our interpretation.

In regards to the phase space density data, we appreciate that the publicly available REPT data is indeed the spin-sectored flux data. More information regarding the processing of the phase space density data has been given in lines 30-49 of the supplementary material (as per Reviewer 2's request also).

Major issues:

Line 73-75: “Here, we study the October 2012 period, where the flux of >7 MeV electrons remained above the background threshold [16] for more than seven days...” I don’t think the 7.7 MeV flux was above background continuously over the time period of these two storms. Can you show evidence of this? The 6.3 MeV channel very much looks like it was above background, but 7.7 MeV looks count-starved during some of this time.

We were referring to the data presented in panel g of Figure 1 where 7.7 MeV flux values above the background threshold are seen during multiple days. However, there are a few orbits of probe A and B during this period where the number of measurements above background threshold of the 7.7 MeV channel is very low, most notably prior to the second enhancement event on the 13th of October. We have changed the wording here accordingly. Lines 86 – 87 now read “*Here we study the October 2012 period, where 7.7 MeV electron flux values, greater than the background threshold [16], were observed during multiple days*”.

Line 88-89: “However, for both storms, Fig 3 (top) more closely resembles local acceleration followed by outwards radial diffusion (Fig 2c).” Not true. During the *recovery period* of the first storm, we can see outward radial diffusion taking place. But during the main phase of the storm, or what you shade gray as the acceleration region, we see evidence of inward radial diffusion (Oct 9 – 10.5).

We have added to lines 104 - 107 to better specify when outward radial diffusion was observed. The sentence now reads “*However, over a time-span of several days, the profiles for both storms shown in Fig 4a more closely resemble local acceleration followed by outward radial diffusion (Fig 3c) during the recovery period*”. Additionally, we have added the following on lines 113 – 114 “*For $\mu = 10,000$ MeV/G, the phase space density peak rapidly shifts across $\sim 0.5L^*$, corresponding to an energy increase from ~ 6 MeV to ~ 7 MeV.*” From the phase space density passes shown in Fig 5, it is unclear whether this shift is due to inward diffusion or a change in the L^* coverage of the local acceleration region.

Examination of the Dst index shown in the last panel of Figure 1 reveals that the recovery period of the first storm began shortly before noon on October 9th. While we agree that the peak in phase space density at $\mu = 8,000$ and 10,000 MeV/G does begin early in the morning of the 9th, in the main phase, we see a phase space density profile with a peaked structure that increases with time indicating local acceleration processes.

We have now separated the phase space density profiles shown into a new, clearer figure (Fig 5) and include in the supplementary information the respective timings and L^* coverage of the two probes (Fig S6). In addition, the resulting profiles from a 1-D radial diffusion VERB model run are also shown to enable a comparison with how the phase space density profile would evolve if radial diffusion were the only process acting at these energies (supplementary section 7). We find that the 1-D model significantly under-estimates observations and does not reproduce the observed phase space density profile.

Line 99-100: “Reeves et al. [12] reported that the $\mu = 3,433$ MeV/G phase space density showed an enhancement between the 8 and 9 October.” You have this data as well – do you also see the enhancement at ~ 3000 MeV/G?

Yes we do. The data processed here for ~ 3000 MeV/G agrees with the results of the Reeves et al. 2013 study. We have now mentioned this in the manuscript (line 134) and make reference to Fig S2a in the supplementary material to show that we also observe a growing peak at this μ , in agreement with previous work.

Line 109: “After the first phase space density enhancement, the peak moved to larger L^* .” What

does this mean? What first enhancement? If you look carefully at Fig 3b-c, you can see that the peak of PSD moves inward during the time shown. You don't show the radial profiles after early on 10 Oct, so it's difficult to see what happens after that but I suspect the peak broadens instead of really strongly peaking at higher L^* , such that the one "peak" can be picked out at higher L^* but really it's more of a plateau. Need to see more information – please put in the paper or Supplemental.

Thank you, the wording here was unclear. As the reviewer suggests, following the initial storm, the peak begins to reduce while the phase space density increases at L^* exterior to the peak, resulting in more of a plateau. We have now included phase space density profiles for later times in supplementary material (Fig S3) and have reworded line 155 - 156 to improve clarity.

Line 111-112: "While the maximum phase space density region expands towards higher L^* for all values of μ shown, for $\mu < 6,000$ MeV/G, an enhanced population remains around $L^* = 4$, not observed for higher values of μ ." What exactly are you referring to here? I don't see this feature in any plots. In supplemental figure S7, I see enhanced PSD at $L^*=4$ for $\mu = 5000$ MeV/G, 9000 MeV/G *and* 12000 MeV/G. What is this difference you are talking about? Please clarify.

Here we were making reference to Fig 4 where, after October 11, the contours for $\mu = 2000, 3000, 4000, \text{ and } 5000$ MeV/G are broader in L^* than the higher μ values, suggesting broader radial profiles for these lower values of μ . We have reworded this in the manuscript (lines 157 - 160) for clarity and changed the <6000 MeV/G to ≤ 5000 MeV/G.

Line 118-122: Again, as for the first double-dip storm, the outward radial diffusion looks like it doesn't play a role until after the main phase, during recovery (here, starting on Oct 15th). In Fig 3d-e, you can see the local peak growing from red to green traces. Then, from green through cyan to the first blue, the peak clearly shifts inward in L^* , indicating *inward* radial diffusion. This is during the acceleration period. After that, you get some complicated dynamics where the peak falls to lower PSD values, shifts outward, then rises and shifts inward again (final pink to red traces). This figure does indeed show the signature of inward radial diffusion, contrary to the final conclusions of the paper.

We agree that the outward radial diffusion doesn't appear to play a role until Oct 15th and, for clarity, have now specified on lines 174 -175 the date when we first saw outward diffusion in the maximum phase space density contours.

At the start of the second storm, magnetopause shadowing occurs, reducing the phase space density. In the previous version of the manuscript, when examining the phase space density profiles from the Van Allen Probes passes, we were not considering the acceleration starting from this reduced phase space density level (instead considering profiles starting early on October 14). We have now rectified this, and show the phase space density profiles from 04:55 October 13 in Fig 5c and d. To avoid over-cluttering the figure, some of the later passes, which were previously shown in the main manuscript, have been moved to supplementary Fig S5.

The phase space density profiles show a peak growing over multiple passes of Van Allen Probe A and B. Initially this peak is centered around $L^* = 4$ and for the final pass is centered around $L^* = 4.2$. The growing peak signature in the measured phase space density profile is evidence of local acceleration. The peak does also shift inwards at later times, and this may indicate some inward radial diffusion, however, this does not explain the growth in the peak. As for the first storm, VERB-1D runs are provided in Supplementary Fig S18 and show that, by only including radial diffusion, we underestimate the phase space density.

We stress that the main conclusion of this paper is not that inwards radial diffusion does not occur here, but rather that the phase space density profiles during this event show signatures of local acceleration directly acting on electrons up to 7 MeV. This is evidenced by the growing peaks observed in the phase space density profiles. As a result, while a combination of inwards radial diffusion and local acceleration could lead to 7 MeV enhancements, the two processes acting simultaneously is not necessarily a requirement to generate such high energy electrons.

Observations of local acceleration up to these energies have not been previously published. We have added lines 200 to 203 to make it clearer that we are not dismissing inwards radial diffusion as a contribution to 7 MeV (and higher energy) enhancements.

Line 124-126: Absolutely not. I see evidence of inward radial diffusion throughout both storms from the PSD radial profiles. How else do you explain the *inward* motion of PSD peaks?

As discussed above, signatures of local acceleration are observed here. In this sentence, we were referencing the two-step acceleration method that was discussed in the introduction. That the phase space density peaks are observed to grow in-situ demonstrates that two-step acceleration is not the only generation mechanism for these ultra-relativistic electrons. They can also be accelerated locally and signatures of this are seen in the phase space density profiles presented. However, we stress that our results do not dismiss two-step acceleration and we were not intending to imply this. Indeed, if very high energy electron populations are accelerated locally, this provides a higher energy ‘source population’ for radial diffusion. We have reworded the sentence (now lines 191 and 193) and have added to this paragraph (lines 200 – 203) to make it clear that we are not dismissing two-step acceleration but demonstrating that in-situ local acceleration can also explain enhancements at these energies.

Line 130-132: Part of this sentence holds: “Our results demonstrate that local acceleration had a significant effect on radiation belt particles during both of the storms in October 2012” The data shows this. However, the data does not support the last part of that sentence: “[local acceleration is] acting on electrons up to 7 MeV.” There is just no way to disentangle the effects of inward radial diffusion and local acceleration in this set of storms, since you see evidence of both processes happening. It might be possible to disentangle in other storm events.

That a phase space density peak at μ values corresponding to ~ 7 MeV is observed to grow with time is evidence of local acceleration acting at this energy, justifying our conclusion that “[local acceleration is] acting on electrons up to 7 MeV.” We have added to the paper after this sentence (lines 200 - 201) to acknowledge that “*In the radiation belt region, local acceleration introduces radial gradients in phase space density and so is always accompanied by both outwards and inwards radial diffusion.*”

Line 136-137: “occurring only a handful of times during the Van Allen Probes mission.” Not true – it has happened at least a couple dozen times, depending on your threshold for acceleration events. I suggest changing this wording.

Thank you, the wording here was subjective. We have changed the sentence to “occurring far less frequently than enhancements at 1 or 2 MeV” (line 208).

Minor issues:

Line 35-36: Awkward language. “keeps electron’s L^* values” should be something like “electron’s drift within constant L^* ” or similar.

Thank you for the suggestion, this has been corrected.

Line 36: Not sure why “producing” is emphasized; shouldn’t it be “growing” that is the crucial term here?

This has been changed.

Line 54: More awkward language. “Consider Fig 1b” should be something like “As shown in Fig 1b”.

This has been changed.

Line 77-79: I don't understand. Fig 2 is a cartoon schematic, right? Why is it described as using "PSD values within half-an-order of magnitude of the maximum for the period"? Which period – the one studied here? I thought this was just a schematic representation – not using actual data. Maybe you meant to say "Fig 3"? In any case, I think Fig 2 should be described first before moving on to describe Fig 3.

Apologies for the confusion. As Fig 3 (previously figure 2) is a schematic representation of how processes would appear when presented in the format of Fig 4a (previously figure 3), the lead author was attempting to explain the format of Fig 4a, using Fig 3 as a prop. The reviewer is right, this is awkward and has been altered. (Lines 91 – 92.)

Line 89: "outward radial diffusion"

Thank you for noticing this. This has been corrected.

Figures:

- Figure 1 needs to be completely redone – it is poor quality and hard to read labels.

For this figure, the tickmarks, axis labels and color bar have all been altered and enlarged. Additionally we have increased the resolution and saved as a different image type.

- Figure 3 needs to be redone. The L* labels need to be bigger in panel a. Resolution needs to be better. (This is true with all figures and may be an artifact of using PDF in LaTeX. Convert to .jpg or .png first and it should help.)

Thank you for the advice regarding figure file types. In addition to enlarging the text in Figs 1 and 4 (previously figure 3) we have changed the file types for all of the figures and feel that, as suggested, this has indeed improved the resolution. Fig 4 now also includes the Kp index for reference.

Reviewer #2 (Remarks to the Author):

This study, using the Van Allen Probes' measurements of multi-MeV electrons, presents a unique way of analyzing electron phase space densities with a wide range of μ values. The results show that during an intense storm of October 2012, electrons are heated up to ~ 7 MeV by local acceleration. Thus, the authors suggest that the acceleration of ultrarelativistic electrons with energies up to ~ 7 MeV can be caused by local heating alone and a two-step process of local acceleration combining with inward radial diffusion, which is suggested by some previous studies, is not required in accelerating electrons to ultrarelativistic energies.

This topic on the acceleration of ultrarelativistic electrons is under-explored due to the lack of high quality data prior to the Van Allen Probes era. Fully understanding the acceleration of ultrarelativistic electrons will contribute to a more comprehensive understanding of the radiation belt dynamics and thus the magnetospheric physics. It will also shed light on the energy-dependent dynamics of radiation belt particles and will be of great interest to others in this field. This manuscript is concise and well-written. However, I still have some concerns about this study's novelty and methods.

We would like to thank the reviewer for their helpful suggestions and careful review of this work. Their time and advice is greatly appreciated. We are glad that they feel that the topic is of particular interest and really appreciate their comment that the paper is well-written. Each of their concerns has been addressed as described below.

First, as the authors also pointed out in their manuscript, several studies on the acceleration of ultrarelativistic electrons have been carried out in the Van Allen Probes era, including both data analysis and modeling. The storm this study focused on is also a widely-studied one. Specifically, Thorne et al. (2013), using a 2-D diffusion model with chorus wave heating, well reproduced the flux enhancements of electrons with energies up to ~ 7 MeV in the outer radiation belt during this Oct 2012 storm and demonstrated the dominant role of local acceleration in ultrarelativistic electron flux enhancements. This study by Allison et al. shows intriguing observational results confirming these results, however, the novelty may be somewhat compromised due to the previous publications.

One of the storms considered in our manuscript has received attention by several other studies. However, none of these previous studies showed observations of phase space density peaks forming for ~ 7 MeV electrons, focusing instead on lower energies, or explored the event with numerical modeling. While a useful investigative tool, modeling the radiation belts is subject to a number of assumptions, and the results achieved may not always be fully representative of all the processes acting. Indeed, that a number of recent studies are presenting a two-step acceleration theory to explain occurrences of ultra-relativistic electrons (and were published after the Thorne et al., 2013 paper) suggests that formation by solely local acceleration is not the accepted theory.

The Thorne et al., 2013 study used a 2-D diffusion model, at a fixed L^* , to explore the extent of chorus wave acceleration. As the model output agreed well with observations at the same L^* , Thorne et al. inferred that local acceleration was the dominant mechanism. However, they did not show observations of local acceleration acting up to 7 MeV as they did not consider the radial profiles of phase space density to explore various energization mechanisms. Because their model was 2-D, the impact of radial diffusion was not considered. Additionally, the model makes a number of assumptions, including using an assumed plasma density model, using quasi-linear theory, and considering only resonant wave-particle interactions. Interestingly, here we also see the peak growing at a lower L^* ($L^* \sim 4.2$) than that considered in the Thorne et al., 2013 paper ($L^* = 5$).

We have now added additional text to the manuscript (see lines 46 - 50 of the updated paper) to better clarify how the work here differs from the Thorne et al., 2013 study (but the results here complement their work, as the reviewer highlights).

Second, this study focuses on the acceleration of ultrarelativistic electrons up to ~ 7 MeV, which has been also emphasized by the authors in several places in the manuscript (abstract, line 132, etc.). However, the main results regarding the evolution of electron phase space densities focus on $\mu=8000$ MeV/G and $\mu=10000$ MeV/G, $K=0.11$ G1/2Re electrons. It was stated in the manuscript that at these μ and K values the energy of electrons correspond to ~ 6 MeV and ~ 7 MeV at $L^*\sim 4.2$ respectively. However, reading from their Figure S1, these μ values should correspond to ~ 5.5 MeV and ~ 6 MeV respectively, and ~ 7 MeV electrons should correspond to much higher μ values at $L^*\sim 4 - 4.5$ ($>\sim 12000$ MeV/G). It is possible that during some portion of this storm the magnetic field was more distorted and these μ values corresponded to higher energies at $L^*\sim 4.2$; however, the mismatch between texts and Figure S1 is still a concern and may confuse readers.

In the previous version of the supplementary material, there was a mistake on Figure S1. Other than 2,000 MeV/G, the line colors did not match the key given, owing to a duplication of the color red in the key, putting the color indexing out-of-sync for all subsequent μ values. Other than the colors, the lines did however match the μ values given. (Ten values on key, ten lines on plot, running from 2,000 MeV/G – 15,000 MeV.G.) The colors have now been corrected in the updated version. We have also chosen to move this figure into the main body of the manuscript.

The third line from the top corresponds to $\mu = 10,000$ MeV/G and therefore covers energies from ~ 7.3 MeV to 6 MeV between $L^*\sim 4-4.5$. The line corresponding to $\mu = 8,000$ MeV/G shows ~ 6.5 MeV to ~ 5.4 MeV for $L^*\sim 4-4.5$. The manuscript text has also been updated to give the energy range for $L^*\sim 4-4.5$ instead of a single value (Figure caption Fig 4).

Third, the way of showing the spatiotemporal evolution of phase space density maxima for a range of μ values is innovative. However, it still suffers from the similar problem with the traditional way of looking at phase space density radial profile at fixed μ and K : without sufficient spatial coverage and time resolution, the results can be somewhat ambiguous. For example, a local peak could be caused by the local acceleration alone but could also be caused by inward radial diffusion combined with fast loss at high L^* . The authors should at least discuss more about this aspect.

Thank you. We have now added lines 180-190 to the updated manuscript to discuss this point.

Additionally, we have also added two further figures to the supplementary material (Figs S8 and S9) to show the last closed drift shell data and the geostationary flux observations from GOES, which did not observe a flux enhancement consistent with a rapid increase at the outer boundary.

Some other comments/questions:

1. The author pointed out the study by Katsavrias et al. (2019) on the importance of two-step acceleration on the ultrarelativistic electron flux enhancements in the April 2017 storm. However, an early study by Zhao et al. (2018) studied the same storm using both phase space density data analysis and radial diffusion modeling, showed the dominant role of inward radial diffusion in the acceleration of ~ 7 MeV electrons in the center of outer belt, and proposed a two-step acceleration mechanism for ultrarelativistic electron flux enhancements. Since this study by Zhao et al. (2018) focused on $K=0.1$ G1/2Re electrons instead of near-equatorially mirroring electrons with $K\leq 0.03$ G1/2Re in Katsavrias et al. (2019), it is more relevant to this study and should be included as a reference.

Zhao, H., Baker, D. N., Li, X., Jaynes, A. N., & Kanekal, S. G. (2018). The acceleration of ultrarelativistic electrons during a small to moderate storm of 21 April 2017. *Geophysical Research Letters*, 45, 5818–5825. <https://doi.org/10.1029/2018GL078582>.

Thank you for drawing our attention to this publication. Zhao et al, 2018 has now been included as a reference.

2. Line 39 "... above energies of ~ 3 MeV, chorus wave acceleration declines...": since this study mainly focuses on the importance of chorus wave acceleration on ultrarelativistic electron flux

enhancements, it would be helpful to have some discussions regarding the reason of chorus wave acceleration declining for ultrarelativistic electrons and why this does not affect the main result of this study.

We have changed the wording here (now lines 41-42) to say “*current theory indicates that acceleration via resonant interactions with chorus waves tends to decline*”, including a new reference (Horne et al., 2003) and added the following on lines 44-46: “*Variable factors, such as the electron number density and the duration of chorus activity, influence the maximum energy achieved by local acceleration and it is therefore possible that chorus interactions can also be responsible for >3 MeV enhancements*”.

3. As the results of this study heavily rely on the accuracy of phase space density calculation, the method used to calculate phase space densities and phase space coordinates will need to be provided in more detail. For example, how was the interpolation done and what assumptions have been used when interpolate the energy spectrum/pitch angle distribution? These information will be critical for readers to reproduce their results.

Thank you for highlighting this. A new subsection in the supplementary information has now been added to detail how the phase space density values were calculated and interpolated (supplementary section 1.2). Additionally, we have also included a line in the main manuscript, directing the reader to this information (lines 88-90).

4. Figure S8 and S9: It would be better to show the corresponding μ values, 8000 MeV/G and 10000 MeV/G, as in the main manuscript, in order to make a more direct comparison.

In the previous version we had used the same μ values as figure S2 (previously figure S3) for consistency across the same figure type. However, we agree that it would be better to match the main manuscript and Figures S8 (now S22) and S9 (now S23) have been changed accordingly.

Reviewer #3 (Remarks to the Author):

This paper examines radial phase space density profiles from two events in October 2012, examining what acceleration mechanism is responsible for the enhancement of ultra-relativistic electrons during this period. They conclude that for these events, local acceleration is responsible for heating these electrons up to 7 MeV.

Overall, the analysis and visualizations in this paper are appropriate and well done. In addition, the results could represent an important and influential contribution to the community. However, some additional analysis/discussion is needed to support the overall conclusions as currently written.

We are very grateful to the reviewer for their time and suggestions as well as glad to see that they feel that the results are important and analysis appropriate. We really appreciate their review. Below we address each of their comments in turn.

Major Comments:

At present, the paper suggests that the presented observations “unequivocally demonstrates that electrons are heated up to 7 MeV from local acceleration alone”. While the paper clearly demonstrates that local acceleration is active during these events and is likely the dominant mechanism, some additional discussion/analysis is needed to justify that local acceleration is solely responsible for the observed enhancements.

We have now reworded this line in the abstract and clarify throughout the manuscript that we are not stating that radial diffusion plays no part in this event, but rather that the growing peaks are evidence that local acceleration is directly accelerating electrons to these energies without radial diffusion acting as a mediating step. Direct observations of local acceleration at these energies has not been seen before and is therefore the novel finding of this study.

For the 9 October event, the authors state: “Throughout the enhancement, negative gradients were observed at $L^* > 4.5$, indicative of local acceleration.” However, looking at Figure 3b-c, there is a clear enhancement near apogee for the 9-Oct 8:45 and 12:40 passes. This coincides with the large enhancement at $L=4.25$. On a related note, I’m unclear why these enhancements near apogee don’t appear in Figure 3a.

The enhanced phase space density at $L^* \sim 4.8$ appears to originate from two factors. Firstly, the energy and local pitch angle corresponding to the μ and K values also show a sharp change over the region, and L^* changes suddenly (see Figs S11, S12, and S13). Examining the ratio of the observed magnetic field to the model magnetic field shows fluctuations during this time (Fig S15). The B_z component increases, before decreasing slightly during the following pass (Fig S16). It is therefore likely that the magnetic field model does not fully describe the state of the Earth’s field at this point. In particular, notice that all local pitch angles sampled apparently correspond to the same L^* (Fig S13). Owing to limitations in the field model, populations could therefore have been assigned the incorrect values for K and L^* at this location (μ is from the observed field) producing the sharp change in phase space density.

Secondly, the electron flux shows an increase at local pitch angles away from 90 degrees (Fig S17). A comparable flux increase near 90° is not initially observed and the local pitch angle distribution inverts. As, over the region, $K=0.11$ corresponds to local pitch angles between 47.6° and 37.1°, this increase would have affected the phase space density profile.

A combination of these two effects likely produced the sharp phase space density bump observed, but we suggest that it is primarily an artifact of the field model. We have now included a number of new figures in the supplementary material in a new section (section 6) to discuss this feature and illustrate to the reader the origin of the bump.

We have changed our previous wording to “Negative phase space density gradients were observed around $L^* \sim 4.7$ for both values of μ , however, there is a very sharp increase towards the

end of the orbit, outside $L^* \sim 4.8$ (see Fig S10)". On lines 147 – 148 we state "*As it would appear that the enhancement at $L^* \sim 4.8$ is an artifact of the field model, we do not show this in Fig 5a and b*". The reader is also directed to the supplementary information for the additional figures and discussion.

If the rapid change in phase space density does not originate from magnetic field discrepancies, then it is a very interesting observation and raises significant questions as to its origin as the last closed drift shell is approximately only half an L^* away and the spacecraft apogee occurs at >6 MLT. It also appears very rapidly and corresponds to energies >3.5 MeV (from Fig S12).

As a test, we included the increase in the outer boundary condition of the VERB-1D model runs (only radial diffusion) shown in supplementary section 7 (Fig S18), and found that, even with the enhancement, the observed phase space density profile at $L^* = 4$ was not reproduced and is considerably lower than observations.

The increase near $L^* \sim 5$ was not seen in Fig 4 (previously Fig 3) as, to form figure 4, the data had to be gridded in time and L^* before being passed to the contour routine. It is for this reason that we also analyze the phase space density along the Van Allen Probes passes in Figure 5.

On a separate note, the L^* values at which we did not have Van Allen Probes measurements had been shaded in black, and to do this we used the $\mu = 2,000$ MeV/G data. At this value of μ , at high L^* , the corresponding energy sometimes was below 1.8 MeV, and so returned no phase space density observation. As a result, during some times, the black region shown in Figure 4 was actually at lower L^* than it should have been for the higher values of μ . We have corrected this.

Similarly, for the 14-15 October event: The 14-Oct 22:14:19 pass in purple for 10000 MeV/G shows a clear secondary peak at $L^* = 5.1$. This precedes some the growing peaks at lower L^* observed in the subsequent passes (00:07 and 03:25).

We have now separated the original Fig 3 into two figures (Figs 4 and 5) to make the phase space density profiles from the Van Allen Probe passes easier to see. The phase space density profiles for the second storm are shown in Fig 5c and d. Passes are labeled by the times at the start and end of the L^* range shown (rather than the average time - which was used previously) and the L^* at each time for the different passes has been provided in supplementary Fig S7.

At the start of the second storm, magnetopause shadowing occurs, reducing the phase space density. In the previous version of the manuscript, when examining the phase space density profiles from the Van Allen Probes passes, we did not consider the acceleration starting from this reduced phase space density level. We have now rectified this, and show earlier profiles starting from 04:55 October 13 in Fig 5c and d.

The phase space density profiles show a peak growing over multiple passes of Van Allen Probe A and B. Initially this peak is centered around $L^* = 4$ and evolves to be centered at around $L^* = 4.2$. To avoid over-cluttering the figure, some of the later passes, which were previously shown in the main manuscript, have been moved to supplementary Fig S5. These include the three passes mentioned here by the reviewer. In supplementary section 3, we discuss these phase space density profiles shown in Fig S5. Although these passes show a further increase in phase space density, they occurred after the primary phase space density increase.

Without observations at larger L^* , this suggests that the observed enhancements at low L^* could be due (at least in part) to rapid inward transport coupled with an on-off source process at higher L^* . This possibility and any constraints the observations would place on it should be discussed.

Lines 180-190 have been added to the main manuscript to discuss the possibility of rapid enhancements at high L^* followed by fast loss. In section 4 of the supplementary material we have included the L^* throughout the orbits of the two Van Allen Probes (at $K = 0.11 R_E G^{1/2}$) during both storm periods (Figs S6 and S7) and consider the time restrictions therefore imposed on any potential on-off source process. Additionally, in section 5 of the supplementary material we have provided the last closed drift shell information and geostationary observations from GOES-13, 14, and 15. The GOES satellites did not observe rapid flux enhancements consistent with an on-off source

population, and instead showed changes in accordance with L^* variations.

Minor Comments:

- Figure 2: This figure is a very nice visualization of the different acceleration mechanisms. My only minor suggestion would be to consider having the local acceleration peak appear at the same L^* in each panel. Relative to panel 1b, panel 1d appears at larger L^* and panel 1c at smaller L^* . If this makes the plots more difficult to read, this suggestion can be ignored.

Thank you very much. We did strongly consider this alteration to Fig 3 (previously Fig 2) but feel that, for panel d), having the local acceleration initially at higher L^* (where the corresponding energy of a specific μ and K would be lower and therefore easier to reach via local acceleration) before the inward radial diffusion, is probably more representative of what is likely to be observed. As a result, we would prefer to have the local acceleration peak at the L^* values previously shown. We have, however, now remade this figure and also adjusted the resolution as per Reviewer 1's suggestion. The figure is similar to before but is clearer and of higher quality.

- Figure 3: Marking these intervals in some way to show inbound vs. outbound passes would be useful.

Thank you for the suggestion. We agree and have switched to using solid and dotted lines for probe A and B respectively and now use left pointing arrows for inbound and right pointing arrows for outbound. The text has been changed accordingly and we feel that the resulting discussion is now easier to follow.

Reviewers' comments:

Reviewer #1 (Remarks to the Author):

Review #2 of Local Heating of Radiation Belt Electrons to Ultra-relativistic Energies by Allison & Shprits

Having reviewed the author responses and changes to the manuscript, I unfortunately still have great concerns about the conclusions of this paper. Overall, I don't see that the data here supports the conclusions unambiguously. Nowhere in the manuscript do you say that inward radial diffusion was clearly acting over the day of Oct 9th on higher energies, although it is very evident in the data (see point 2 below), so it appears that you are actively avoiding this discussion. I think this topic is extremely timely and of great interest to the community but I do not see the evidence in this event as presented. It may be possible to find another event which unambiguously shows local acceleration directly leading to 7.7 MeV electron enhancements (very clear growing peak signatures – much more clear than shown here), and I encourage authors to look for an event like this.

In summary, I do not recommend publication of the manuscript. It is an interesting analysis of the storm event, but without robust support of the stated conclusion, it is not new enough to constitute publication in this kind of journal.

Major issues still unresolved:

1. In the first review, I asked that you show evidence that the 7.7 MeV electron channel was above background. It turns out it was at the 1 or 2 count level for the beginning of the Oct 8-9 period. The same is true for some of the lower energy channels. Channels 5-7 appear to be below background until the pass starting at ~2 UT on 10/9, and Channels 6-7 are still below background until the inbound pass starting at ~6 UT on 10/9 and the next outbound pass starting at ~11 UT on 10/9, respectively. However, for the Oct 13-14 period, all channels were above background levels and that data is viable.

The concern that comes with this surrounds the validity of the PSD profiles for the Oct 8-9 period. For the first part of this period, the flat PSD curves (dark blue and black in your plots) are meaningless. The data used to derive PSD is not robust. The first light blue curve is also not representative, since the values at low L will be from those data points below background (you can see the PSD values are very close to the dark blue and black curves). The second light blue curve is the most valid point to start the analysis – here, all data points are likely above background and we

can trust the shape of that profile. Except, when I look at the publicly available PSD, I see PSD values are not the same as what you show. The magnitudes are very similar, but the trend doesn't always look quite the same. This can be true depending on the different methods used to calculate PSD, but in this case I would say it is not clear that there are growing peaks in either storm case. In fact, from your Oct 13-14 plots, I don't see growing peaks at all. Green and blue curves look like positive or flat gradients – the peak is only evident in the red (maybe orange for 8000 MeV/G) traces, at which time the maximum magnitude has stopped increasing.

You can see this in the Supplemental information as well. Look at your Fig S1, 8000 and above panels. You see no growing peak – or else you would see a step by step increase in color at some mid-L value. You can check the public PSD data on the RBSP Gateway for comparison:
<http://rbspgateway.jhuapl.edu/PSD>

2. This next issue is in line with the previous one, but directly relates to my original second major issue (called "Line 88-89" in the first review). I still see clear evidence of inward radial diffusion from Figure 4. It's hard to see all the μ values clearly, but at least for the 10000 MeV/G and 8000 MeV/G values (the two red layers on top), there is very clearly an inward movement of the peak before a flattening and/or outward motion. I'm including an annotated figure with my review to show the period I'm referring to.

As for the radial diffusion modeling, this can be tuned according to what diffusion rates are used and we know that many events do not follow the K_p parameterized values but have event-specific rates that can differ greatly from storm to storm. Showing one run of a 1-d diffusion model does not support the conclusion that local acceleration was the main driver here. Other groups have successfully modeled all Van Allen Probes observations using radial diffusion alone (see Mann and Ozeke, 2016, JGR) – it just depends on the rates you decide to use.

3. I suggest plotting RBSP-B PSD profiles as well, since A shows some complicated dynamics during both of these storms, while B seems to catch the belts during the more stable intervals (overall). I don't think this will strengthen the argument you are putting forth, but it may be worth looking at more closely.

4. In your newly added Supplemental material, you say you used 5 minute averaged REPT data (Line 33-34). How are you using averaged data to get individual sector information for pitch angle direction? Or do you mean that you are averaging after calculating PSD? Or are you averaging within each pitch angle bin by 5 minutes?

Related to that, in your method you say that you interpolate PSD to a target value of K, but don't mention a model pitch angle distribution. Is it a linear interpolation from one K-value to the next to ascertain the point in space that matches your target K? If so, this may introduce errors since the pitch angle bins can be very under-sampled (even somewhat close to 90 degrees depending on angle of spacecraft to Bfield). Usually a model pitch angle distribution is used to account for this (which I know has its own problems). What happens if you instead target a certain Mu value and base your interpolation on that? Does it change your results much? The reason I asked is based on the discrepancy between your PSD profiles and the ones I have access to on the Gateway, which may have been derived in a different way.

Reviewer #2 (Remarks to the Author):

The authors have addressed some of my comments. However, I still have some concerns regarding the responses to my third major comment. The authors have added some discussions (lines 180 – 190) and also figures (Figs S8 and S9) to the manuscript to address the spatiotemporal ambiguity in the phase space density radial profile. Specifically, it was stated that “Furthermore, the last closed drift shell location is at $L^* \sim 5.5 - 6.5$ for the first storm and $L^* \sim 6 - 6.5$ for the second (Fig S8). The minimum energy of the source population that would be required for the observed 7 MeV enhancements is >3 MeV (Fig 2) and it is unclear where this would originate.” However, Fig 2 does not give the electron energy at $L^* > 5.6$, making it hard for readers to know what the corresponding energy of the potential source population around GEO would be. Also, the statement that “it is unclear where this would originate” is hard to understand – is it saying that there are not so many high energy electrons at the high L region to act as the source population for >7 MeV electrons in the center of the outer belt? In addition, it was stated that “These additional observations suggest an absence of a rapidly appearing source population for radial diffusion during both events.” However, GOES data in Fig S9 does show some flux enhancements of >2 MeV electrons especially during the first storm; and because of GOES's integral channel and often steep-falling electron energy spectrum in the radiation belts, GOES measurements may not show the actual corresponding source populations of >7 MeV electrons in the center of the outer belt. I would suggest to change the wording and point out this limitation of current in situ measurements, which are not sufficient to infer the source populations of ultrarelativistic electrons at the high L region.

Another minor comment: a manuscript recently published by Zhao et al. suggested the dominant role of inward radial diffusion in the acceleration of ultrarelativistic electrons through statistical analysis on 19 electron flux enhancement events during the Van Allen Probe era. While their results

may not necessarily contradict this study as they focused on the general statistics while this study focuses on two specific storms, it is beneficial to discuss their results in the context of this study.

Reference: Zhao, H., Baker, D. N., Li, X., Malaspina, D. M., Jaynes, A. N., & Kanekal, S. G. (2019). On the acceleration mechanism of ultrarelativistic electrons in the center of the outer radiation belt: A statistical study. *Journal of Geophysical Research: Space Physics*, 124, 8590–8599.

<https://doi.org/10.1029/2019JA027111>.

Reviewer #3 (Remarks to the Author):

This paper addresses the acceleration of ultra-relativistic

electrons using phase space density observations from the Van Allen Probes. This is an important and relevant result and should be an impactful contribution to the community.

The authors revisions have adequately all of my concerns and I am happy to recommend publication.

Response to reviewers

We thank all of the reviewers for their time. Their comments are addressed below in blue. Unless otherwise stated, line numbers correspond to the model up-to-date version of the track-changes manuscript.

Reviewer #1

Having reviewed the author responses and changes to the manuscript, I unfortunately still have great concerns about the conclusions of this paper. Overall, I don't see that the data here supports the conclusions unambiguously. Nowhere in the manuscript do you say that inward radial diffusion was clearly acting over the day of Oct 9th on higher energies, although it is very evident in the data (see point 2 below), so it appears that you are actively avoiding this discussion.

As requested in the previous review, we did add discussion of inward radial diffusion to the revised manuscript – see lines 102-103 (previously on lines 113 – 114 of the tracked changes document), quantifying the energy change produced, which is only ~600keV, less than 10% of the energy of the observed 7.7 MeV electrons. We included a discussion on lines 186 -189 (previously on lines 200 – 201 of the tracked changes document) highlighting that inward radial diffusion is always active alongside local acceleration because local acceleration introduces radial gradients in phase space density. We also had a section in the supplementary material discussing inward diffusion (supplementary section 10). Hopefully these discussions that were overlooked by the reviewer in the previous review will satisfy their questions.

I think this topic is extremely timely and of great interest to the community but I do not see the evidence in this event as presented. It may be possible to find another event which unambiguously shows local acceleration directly leading to 7.7 MeV electron enhancements (very clear growing peak signatures – much more clear than shown here), and I encourage authors to look for an event like this.

We are glad to see that the reviewer finds this topic interesting, and we take their suggestion that we should continue to analyze the phase space density profiles surrounding other 7.7 MeV enhancement events. We believe that the profiles shown here are conclusive and note that reviewer #2 and reviewer #3 also did not raise concerns regarding the interpretation. Here we have included an annotated version of figure 5 to show where we see growing peaks:

In summary, I do not recommend publication of the manuscript. It is an interesting analysis of the storm event, but without robust support of the stated conclusion, it is not new enough to constitute publication in this kind of journal.

We thank the reviewer for their time and review and appreciate the comment that they find the analysis interesting. We are sorry to see that they do not recommend publication. We consider the results presented in the manuscript to be clear and have addressed their points below.

Major issues still unresolved:

1. In the first review, I asked that you show evidence that the 7.7 MeV electron channel was above background. It turns out it was at the 1 or 2 count level for the beginning of the Oct 8-9 period. The same is true for some of the lower energy channels. Channels 5-7 appear to be below background until the pass starting at ~2 UT on 10/9, and Channels 6-7 are still below background until the inbound pass starting at ~6 UT on 10/9 and the next outbound pass starting at ~11 UT on 10/9, respectively. However, for the Oct 13-14 period, all channels were above background levels and that data is viable. The concern that comes with this surrounds the validity of the PSD profiles for the Oct 8-9 period. For the first part of this period, the flat PSD curves (dark blue and black in your plots) are meaningless. The data used to derive PSD is not robust. The first light blue curve is also not representative, since the values at low L will be from those data points below background (you can see the PSD values are very close to the dark blue and black curves).

For the storm time the instruments were above background. Black and dark blue curves may have been close to the background level prior to the first event, but these have been included to illustrate the upper bound of the phase space density prior to the enhancement. A phrase to this effect has now been added to the manuscript on lines 108 - 110.

The second light blue curve is the most valid point to start the analysis – here, all data points are likely above background and we can trust the shape of that profile. Except, when I look at the publicly available PSD, I see PSD values are not the same as what you show. The magnitudes are very similar, but the trend doesn't always look quite the same. This can be true depending on the different methods used to calculate PSD, but in this case I would say it is not clear that there are growing peaks in either storm case.

In the main manuscript we used the TS07 field model for the K and L invariants and the local measurement of the magnetic field for the calculation of mu. The profiles corresponding to the TS07 field model do not load in the gateway. The gateway returns the following figure:*

So here we have included the phase space density profiles from the publicly available gateway with the TS04 field model instead. Note that growing peaks are still observed.

The gateway does not provide documentation on how the phase space density was calculated, so it is difficult to comment on whether there is a differing approach to how the data was processed. Note that we have made our script for interpolating the phase space density to a target μ and K pair available to the reviewers, as well as the PSD values, so that you can see our processing steps.

In fact, from your Oct 13-14 plots, I don't see growing peaks at all. Green and blue curves look like positive or flat gradients – the peak is only evident in the red (maybe orange for 8000 MeV/G) traces, at which time the maximum magnitude has stopped increasing.

During the first storm, as shown in Figure 5 (and the annotated figure 5 at the start of this document):

- negative phase space density gradients are observed between $L^* = 4.6$ and $L^* = 4.9$ throughout the enhancement, at both $\mu = 10,000$ MeV/G and $\mu = 8,000$ MeV/G.
- During this time, the peak phase space density increases from levels near the background threshold of the instrument to 4×10^9 $c^3 \text{cm}^{-3} \text{MeV}^{-3}$ for $\mu = 8,000$ MeV/G and to 8×10^{10} $c^3 \text{cm}^{-3} \text{MeV}^{-3}$ for $\mu = 10,000$ MeV/G.

This is a growing peak.

During the second storm, the initial phase space density is higher than for the first storm event, as it has not been fully depleted following the previous enhancement.

Figure 5 (and the annotated figure 5 at the top of this document) shows that:

- The final profile has a peak in phase space density at $L^* = 4.2$ with a phase space density value greater than the pre-storm levels
- Profiles show negative phase space density gradients between $L^* = 4.6$ and $L^* = 4.9$ throughout the enhancement.
- The light blue inbound pass of Van Allen Probe A (solid line) is the exception, which shows a positive gradient in this L^* range. However, the phase space density at the largest L^* during this pass is still lower than the value of the final peak, and so inward diffusion from this point cannot explain the resulting enhancement. Also, the pass immediately after (dotted outbound light blue pass from Van Allen Probe B) again shows negative phase space density gradients

between $L^* = 4.6$ and $L^* = 4.9$ (as do all subsequent passes). The change in phase space density between the solid light blue and dashed light blue passes is small for $\mu = 8,000$ MeV/G and not seen for $\mu = 10,000$ MeV/G.

The negative phase space density gradients between $L^* = 4.6$ and $L^* = 4.9$ and the increase in the level of phase space density constitutes a growing peak.

You can see this in the Supplemental information as well. Look at your Fig S1, 8000 and above panels. You see no growing peak – or else you would see a step by step increase in color at some mid-L value. You can check the public PSD data on the RBSP Gateway for comparison: <http://rbspgateway.jhuapl.edu/PSD>

Figure S1 shows the phase space density for all of the different values of μ on the same color scale. We presented the data in this way for this supplementary figure to enable a comparison of the phase space density magnitude between the μ values. As a result, for the larger values of μ , where the phase space densities are much smaller, it is difficult to assess the scale of the changes.

The line plots of phase space densities shown in figure 5, and throughout the supplementary material, enables the change in phase space density to be quantified. These line plots show growing peaks.

For the benefit of the reviewer, we include here the $\mu = 8,000$ MeV/G and $\mu = 10,000$ MeV/G panels of Figure S1 (now Figure S2) on a different color scale. An increase in color at some mid-L value can be observed.

2. This next issue is in line with the previous one, but directly relates to my original second major issue (called “Line 88-89” in the first review). I still see clear evidence of inward radial diffusion from Figure 4. It’s hard to see all the μ values clearly, but at least for the 10000 MeV/G and 8000 MeV/G values (the two red layers on top), there is very clearly an inward movement of the peak before a flattening and/or outward motion. I’m including an annotated figure with my review to show the period I’m referring to.

In the previous revision of the manuscript, we included lines 113 -114 (line numbers correspond to the previous track-changes version of the manuscript) to address this comment, acknowledging the slight inward motion of the peaks and quantifying the increase in energy range that this corresponds to. From Figure 2, the motion from $L^* = 4.5$ to $L^* = 4.2$ is ~ 600 keV, less than 10% of the energy of a 7.7 MeV electron. We have now reworded lines 102 -103 (in current version of the manuscript) for clarification.

As for the radial diffusion modeling, this can be tuned according to what diffusion rates are used and we know that many events do not follow the Kp parameterized values but have event-specific rates that can differ greatly from storm to storm. Showing one run of a 1-d diffusion model does not support the conclusion that local acceleration was the main driver here. Other groups have successfully modeled all Van Allen Probes observations using radial diffusion alone (see Mann and Ozeke, 2016, JGR) – it just depends on the rates you decide to use.

While we agree that radial diffusion coefficients can be tuned to give particular results, we stress that for the runs included in the supplementary material we have used the standard parameterized radial diffusion coefficients from Brautigam and Albert, 2000, which have not been tuned, and are driven by the observed values of the Kp index for the period in question. These coefficients have been used in many publications (Glauert et al., 2018; Zhao et al., 2018; Drozdov et al., 2017, etc) and generally reproduce observed dynamics well. Of course, the model runs in the supplementary material are not the main focus of the paper, which instead concentrates on observations, and are included only as an illustration, and do not affect the conclusions. Note that Drozdov et al., 2017 provides a comparison between using both the Brautigam and Albert, 2000 diffusion rates and Ozeke et al., 2014 diffusion rates in a numerical model, and finds that they give similar results.

References

Glauert, S. A., Horne, R. B., & Meredith, N. P. (2018). A 30-year simulation of the outer electron radiation belt. *Space Weather*, 16, 1498–1522. <https://doi.org/10.1029/2018SW001981>

Zhao, H., Baker, D. N., Li, X., Jaynes, A. N., & Kanekal, S. G. (2018). The acceleration of ultrarelativistic electrons during a small to moderate storm of 21 April 2017. *Geophysical Research Letters*, 45, 5818–5825. <https://doi.org/10.1029/2018GL078582>

Drozdov, A. Y., Shprits, Y. Y., Aseev, N. A., Kellerman, A. C., and Reeves, G. D. (2017), Dependence of radiation belt simulations to assumed radial diffusion rates tested for two empirical models of radial transport, *Space Weather*, 15, 150–162, doi:10.1002/2016SW001426.

Ozeke, L. G., I. R. Mann, K. R. Murphy, I. Jonathan Rae, and D. K. Milling (2014), Analytic expressions for ULF wave radiation belt radial diffusion coefficients, *J. Geophys. Res. Space Phys.*, **119**, 1587–1605, doi:10.1002/2013JA019204.

3. I suggest plotting RBSP-B PSD profiles as well, since A shows some complicated dynamics during both of these storms, while B seems to catch the belts during the more stable intervals (overall). I don't think this will strengthen the argument you are putting forth, but it may be worth looking at more closely.

There is a misunderstanding here, we do show data from probe A and B as stated in the text and figure captions.

4. In your newly added Supplemental material, you say you used 5 minute averaged REPT data (Line 33-34). How are you using averaged data to get individual sector information for pitch angle direction? Or do you mean that you are averaging after calculating PSD? Or are you averaging within each pitch angle bin by 5 minutes? *Each pitch angle and energy bin is converted to phase space density by dividing by the particle momentum squared. We then average the phase space density within each energy and pitch angle channel into 5 minute intervals. The resulting PSD grid is then interpolated to the targeted mu and K values. The PSD does not vary much over the 5 minute time frame.*

Related to that, in your method you say that you interpolate PSD to a target value of K, but don't mention a model pitch angle distribution. Is it a linear interpolation from one K-value to the next to ascertain the point in space that matches your target K?

As discussed on line 43-45 of the supplementary material, we linearly interpolate the log of the phase space density to the target value of K.

If so, this may introduce errors since the pitch angle bins can be very under-sampled (even somewhat close to 90 degrees depending on angle of spacecraft to Bfield). Usually a model pitch angle distribution is used to account for this (which I know has its own problems).

As discussed on lines 46 – 49 of the supplementary material, we do not use extrapolation and do not use a model pitch angle distribution. Although a model pitch angle distribution was not used, as shown below, using one has a very limited effect on the output. This analysis has been added to the supplementary material as Figure S1 and lines 49 - 51.

The plot shows the phase space density profile at $\mu = 8,000$ MeV/G and $K = 0.11 R_E G^{1/2}$, from Van Allen Probe A on an example pass (15th October 2012 02:25:00 to 05:20:00). The linear interpolation of the logged phase space density is used as is in the paper (red line) and we also used a $psd(\alpha) = A \sin(\alpha) + C$ pitch angle distribution between the pitch angles surrounding the target value of K (blue curve). The two curves are mostly indistinguishable. This Figure has been added to the supplementary information.

As an additional test, we again compute the phase space density for the above example pass using all of the pitch angle bins (black line) and also using a reduced dataset where we had removed every other pitch angle bin (blue line). This reduces the number of available pitch angles from 9 to 5. As you can see from the below figure, reducing the number of pitch angles has a limited effect on the phase space density output.

The plot shows the phase space density profile at $\mu = 8,000$ MeV/G and $K = 0.11 R_E G^{1/2}$, from Van Allen Probe A on an example pass (15th October 2012 02:25:00 to 05:20:00). As in the figure above. The phase space density is calculated using all pitch angle bins (black line) and also calculated using a subset of the data comprising every other pitch angle bin (blue line).

In the paper, the same method for determining the phase space density at a particular μ and K has been used as was used in the Reeves et al., 2013 study, even down to the 5 minute averaging. See supplementary information of that reference for details. **Reference:** Reeves, G. D., H. E. Spence, M. G. Henderson, S. K. Morley, R. H. W. Friedel, H. O. Funsten, D. N. Baker, S. G. Kanekal, J. B. Blake, J. F. Fennell, S. G. Claudepierre, R. M. Thorne, D. L. Turner, C. A. Kletzing, W. S. Kurth, B. A. Larsen, and J. T. Niehof (2013), Electron acceleration in the heart of the Van Allen radiation belts, *Science*, 341 (6149), 991{994, doi:10.1126/science.1237743.

What happens if you instead target a certain μ value and base your interpolation on that? Does it change your results much? The reason I asked is based on the discrepancy between your PSD profiles and the ones I have access to on the Gateway, which may have been derived in a different way.

We do interpolate between energy measurements to a specific μ . In the analysis the phase space density shown is always a specific μ value and a specific K value. It is unclear what reviewer means here.

The mu values we present are calculated with the locally measured magnetic field model as described in lines 35-38 of the supplementary material. Note that we made the phase space density data available to the reviewers, along with the Matlab script for interpolating the data to the target mu and K pair.

Reviewer #2

The authors have addressed some of my comments. However, I still have some concerns regarding the responses to my third major comment.

We thank the reviewer for his/her second review. We really appreciate their time and useful comments.

The authors have added some discussions (lines 180 – 190) and also figures (Figs S8 and S9) to the manuscript to address the spatiotemporal ambiguity in the phase space density radial profile. Specifically, it was stated that “Furthermore, the last closed drift shell location is at $L^* \sim 5.5 - 6.5$ for the first storm and $L^* \sim 6 - 6.5$ for the second (Fig S8). The minimum energy of the source population that would be required for the observed 7 MeV enhancements is >3 MeV (Fig 2) and it is unclear where this would originate.” However, Fig 2 does not give the electron energy at $L^* > 5.6$, making it hard for readers to know what the corresponding energy of the potential source population around GEO would be.

We agree, the number given was inferred from the trends shown in the figure. We have now rephrased this line in the manuscript so that it reads “The trends shown in Figure 2 then suggest that the minimum energy of the source population required for the observed 7 MeV enhancements is likely to be several MeV” as it is difficult to provide the exact energy value. See line 164-166. This is included to justify why we look at the >2 MeV channel of GOES.

Also, the statement that “it is unclear where this would originate” is hard to understand – is it saying that there are not so many high energy electrons at the high L region to act as the source population for >7 MeV electrons in the center of the outer belt?

Here we were commenting that near the last closed drift shell, particles can be lost very quickly and there is a limited time for local acceleration to act. Due to limited observations of the region it is difficult to understand the dynamics there. This phrase has now been removed as it is not directly needed for the conclusions of our study. Instead, we lead into the discussion of the GOES observations.

In addition, it was stated that “These additional observations suggest an absence of a rapidly appearing source population for radial diffusion during both events.” However, GOES data in Fig S9 does show some flux enhancements of >2 MeV electrons especially during the first storm; and because of GOES’s integral channel and often steep-falling electron energy spectrum in the radiation belts, GOES measurements may not show the actual corresponding source populations of >7 MeV electrons in the center of the outer belt. I would suggest to change the wording and point out this limitation of current in situ measurements, which are not sufficient to infer the source populations of ultrarelativistic electrons at the high L region.

Because of the large pitch angle range sampled by the GOES satellites, and the fact that L^ is pitch angle dependent, there is some ambiguity as to the exact L^* of the populations measured by GOES. To account for this, we showed the L^* values for a number of different local pitch angles at the GOES location alongside the GOES flux plots. The L^* range covered by a single GOES measurement can extend over $\sim 1 L^*$. Note that the changes in the flux correspond to transitions in this sampled L^* range and so are likely a reflection of changes in the population that is being measured. Note also*

that the flux recorded by GOES is higher in the days following the events than it was during the acceleration window at $L^ \sim 4$. For this period, that is consistent with locally accelerated source that diffuses outwards.*

We have now changed the wording here, as requested, and include a discussion regarding the limitation of current in situ measurements as well as the integral energy channels. See lines 168 – 177 which now read: “When considered alongside the changes in the L^ measured, these additional observations do not suggest a rapidly appearing source population for radial diffusion during either event. Furthermore, the GOES flux is higher in the days following the acceleration occurring at $L^* \sim 4$ than it was during the enhancement period, consistent with a locally accelerated source that diffuses outwards. However, we note that these are again in situ measurements and therefore also subject to certain spatial limitations. The GOES spacecraft may not have been appropriately situated to measure a source population during the period. Additionally, because radiation belt electrons often show steep falling energy spectra, the integral energy channels may not necessarily show the source populations of >7 MeV electrons in the outer belt.”*

Another minor comment: a manuscript recently published by Zhao et al. suggested the dominant role of inward radial diffusion in the acceleration of ultrarelativistic electrons through statistical analysis on 19 electron flux enhancement events during the Van Allen Probe era. While their results may not necessarily contradict this study as they focused on the general statistics while this study focuses on two specific storms, it is beneficial to discuss their results in the context of this study. Reference: Zhao, H., Baker, D. N., Li, X., Malaspina, D. M., Jaynes, A. N., & Kanekal, S. G. (2019). On the acceleration mechanism of ultrarelativistic electrons in the center of the outer radiation belt: A statistical study. *Journal of Geophysical Research: Space Physics*, 124, 8590–8599. <https://doi.org/10.1029/2019JA027111>.

We agree that this is beneficial to highlight that our results do not conflict with this previous work and have now included a discussion of the results from the above reference in lines 190-196 of the manuscript. The above reference has been added to the reference list.

Reviewer #3

This paper addresses the acceleration of ultra-relativistic electrons using phase space density observations from the Van Allen Probes. This is an important and relevant result and should be an impactful contribution to the community.

The authors revisions have adequately all of my concerns and I am happy to recommend publication.

We thank the reviewer for their time and review of the manuscript and are happy to see that they recommend publication.

REVIEWERS' COMMENTS:

Reviewer #1 (Remarks to the Author):

Comments are included in attached PDF (since it contains images).

Reviewer #2 (Remarks to the Author):

The authors have addressed my concerns and I recommend the publication of this manuscript in the present form.

Additional comments of Reviewer #2 during consultation:

1. I agree with the reviewer that the inward motion of the peak in PSD profile during the first storm could be the result of inward diffusion; however, it could also be due to the local acceleration with a shifting source. The authors did mention this in the revised manuscript as "For $\mu = 10,000$ MeV/G, the phase space density peak rapidly shifts across $\sim 0.3L^*$, which from Fig 2 constitutes an energy change of ~ 600 keV, less than 10% of the energy of a 7 MeV electron." I agree that it is better to be clearer about this inward shift and the possibility of other interpretation, and I would also suggest the authors to revise the second half of this sentence, as pointed out by the reviewer, this statement is correct but may be misleading – those 7 MeV electrons could be accelerated from ~ 6.5 MeV or so. With that being said, I do think that the initial enhancements of high energy electrons during the first storm indeed show a local PSD peak, suggesting the dominant role of local acceleration during this event.

2. Uncertainties always exist when using PSD evolution to infer the underlying physical mechanism, which naturally come from the always limited spatial/temporal resolution of measurements. The novel method of presenting PSD evolution in this study does help reduce such uncertainties, and to me the results do support the dominant role of local acceleration during this event. The authors did also discuss the potential ambiguities in the last paragraph of page 9. It is also worth noting that some high-profile papers, such as Reeves et al. (2013) Science paper, also used the growing PSD peak to infer the dominance of local acceleration though with such type of uncertainties. Overall, I do think that this study highlights the important role of local acceleration on ultra-relativistic electron energization, and demonstrates that a two-step process of local acceleration combining

with inward radial diffusion, which is suggested by some previous studies, is not required in accelerating electrons to ultrarelativistic energies.

Response to reviewers

Reviewer #1:

Review #3 of Local Heating of Radiation Belt Electrons to Ultra-relativistic Energies by Allison & Shprits

It seems that my perspective has been over-ruled by the other 2 reviewers, so I will not request further revisions. I do, however, want to make several final observations for the record.

We thank the reviewer for their time and input. In response to the concerns raised throughout the three reviews we have included additional data from GOES, 1-D model runs, and reformulated parts of our analysis, all of which has improved the manuscript and we therefore thank the reviewer for their comments. Although we may not agree on some aspects of this study, we do appreciate the time and care taken in evaluating our work. We are pleased to see that, upon consultation regarding the points raised in your review, the second reviewer also agrees with our interpretation.

1. Following the introduction and setup (after Figure 3) there is still no explicit mention of inward radial diffusion contributing to the Oct 8-9 event, although I have asked for this to be part of the manuscript and it is clearly seen in the L^* vs. time plot of PSD. (I've created a different annotated snippet of the figure, as shown below.) The closest mention is the following sentence on Lines 110-111: "The broadening of the phase space density peak is likely a result of both inward and outward radial diffusion, ..." but that's not what I've pointed to in my annotation below, or in the text in prior reviews. There is no outward and inward motion shown equally here in this figure (white arrow approximating the evolution over time). There is a net inward diffusion, which is not commented on in the manuscript and so only leads me to believe that the authors are taking a narrow (or inaccurate) interpretation of the data. I find this troubling, and wanted to note this fact here.

We believe that the confusion here comes from misunderstanding that this is not a plot of phase space density in L^* and time, but rather contours indicating the L^* coverage of phase space density enhancements (specifically where PSD was within a factor of 5 of the maximum value for the period). We have rewritten part of the figure caption of Figure 4, to better explain what is shown, and reworded the discussion of the schematic Figure 3 (where we introduce the format of Figure 4) to improve clarity (see lines 92 – 93). The snippet above shows an L^* change of the dark red contour, which signifies an inward motion of the phase space density peak. The L^* change of the peak location was discussed in the revised manuscript, as acknowledged in the 4th item of your review. This can be due to radial diffusion or from the inward motion of the local acceleration region – or even a combination of both. We noted this shift in the manuscript and used Figure 2 to help quantify the energization that it could account for. Following your updated comment and the

suggestion made by reviewer 2, we have now reworded this point in the main manuscript. See lines 109 – 114 of the tracked changes document.

2. Again, I think it's futile to request a rebuttal, but I want to point out that the PSD profiles you have shown in Figure 5 are not unambiguously showing growing PSD peaks at a specific location. First, please take out the black and dark blue traces – they are related to near-background levels of counts and thus will not reveal anything real about the PSD values OR gradients during this time.

As stated previously, we would like to keep the black and dark blue traces in the plots for the first storm as they illustrate that, prior to the enhancement, the phase space density was at a much lower level, with the values plotted signifying an upper threshold. Following your previous review we had added this discussion to the manuscript to discuss this.

You are left with the cyan through red traces. Now, look carefully at just these traces and you will be hard-pressed to show actual growing peaks at a constant L^* location. All of the panels show an overall increase in PSD with time, true, but most of these peaks shift from higher L^* to lower L^* and back again from orbit to orbit, while also growing in magnitude.

Some orbits show negative PSD gradients (outside of the peak PSD) while others show flat profiles, which then change again to negative on the next orbit. These are not clearly growing peaks at (for example) $L^*=4.2$, indicating local acceleration occurring centered on this location. Instead, these profiles point to complicated dynamics of PSD and fluctuating PSD levels (see panel c, where from yellow-green to orange to red, the PSD level goes up then down then up again at around $L^*=4.3$). Fluctuating PSD are commonly observed during storm times and therefore make it very difficult to draw simple conclusions during the main phase of the storm – something you are attempting to do here. I do not agree that the data support the main conclusions made by this paper.

For $L^* = 4 - 4.4$, the phase space density grows with time whilst negative phase space density profiles are seen at larger L^* . It is this which we interpret as a growing peak as it is unclear how, without a very rapid on-off source mechanism operating on timescales smaller than observation rate of the probes (which we mention in the manuscript and look for using GOES data), radial diffusion would allow the phase space density at $L^* = 4$ to increase while measurements exterior showed negative phase space density gradients. Furthermore, by considering the phase space density over a range of μ values and overlaying the enhancement regions, we find a picture which, by our interpretation (and seemingly also reviewer 2 and 3's), is consistent with a local acceleration process acting over a broad energy range, up to ultra-relativistic energies. For the various μ values, we find a peaked phase space density structure centered on a similar range of L^* . Much of the variations that you discuss are likely to be a result of temporal and special limitations of the spacecraft. For example, in Figure 5a and b, the dotted cyan line is less peaked than the surrounding profiles. As this is an outbound pass of spacecraft B, it is likely that acceleration occurred on a time frame comparable to the Van Allen Probe orbit.

3. I wonder why the Gateway does not have the results for the TS07 magnetic field model available? I will inquire.

Thank you.

4. In your reply to my review, you write "In the previous revision of the manuscript, we included lines 113 -114 (line numbers correspond to the previous track-changes version of the manuscript) to address this comment, acknowledging the slight inward motion of the peaks and quantifying the increase in energy range that this corresponds to. From Figure 2, the motion from $L^* = 4.5$ to $L^* = 4.2$ is ~ 600 keV, less than 10% of the energy of a 7.7 MeV electron. We have now reworded lines 102 -103 (in current version of the manuscript) for clarification." But nowhere in lines 102-103 do you mention inward motion or radial diffusion. And I'm not sure why it is significant to say that the motion from 4.5 to 4.2 constitutes an energy increase of 600 keV, less than 10% of the 7.7 MeV electron. Of course the electrons would not increase from 10 keV to 7.7 MeV in this

inward motion, but they will come from slightly lower energies to become 7.7 MeV, thus first appearing as 7.7 MeV after the lower energies are seen.

The reply here is the same as the response to comment 1 as it is addressing the same point on the shift in L^* . As mentioned above, we have reworded this point in the main manuscript following your comments and the suggestion made by reviewer 2. We have also revised the second half of the sentence for clarity. See lines 109 – 114 of track-changes document.

We discuss the energies here, as in the context of the study, which is focused on energization mechanisms, it is important to quantify the energy change introduced if the populations did undergo an inward radial motion of $\sim 0.3 L^*$. We have now reformulated the part of the sentence addressing the energies to make this clearer.

Reviewer #2:

The authors have addressed my concerns and I recommend the publication of this manuscript in the present form.

We thank the reviewer for their time and help in evaluating this manuscript. Their comments were insightful and have improved the paper. We thank them again for additionally taking the time to provide consultation regarding the points made by reviewer 1. We are pleased to see that they recommend publication.

Additional comments of Reviewer #2 during consultation:

1. I agree with the reviewer that the inward motion of the peak in PSD profile during the first storm could be the result of inward diffusion; however, it could also be due to the local acceleration with a shifting source. The authors did mention this in the revised manuscript as “For $\mu = 10,000$ MeV/G, the phase space density peak rapidly shifts across $\sim 0.3L^*$, which from Fig 2 constitutes an energy change of ~ 600 keV, less than 10% of the energy of a 7 MeV electron.” I agree that it is better to be clearer about this inward shift and the possibility of other interpretation, and I would also suggest the authors to revise the second half of this sentence, as pointed out by the reviewer, this statement is correct but may be misleading – those 7 MeV electrons could be accelerated from ~ 6.5 MeV or so. With that being said, I do think that the initial enhancements of high energy electrons during the first storm indeed show a local PSD peak, suggesting the dominant role of local acceleration during this event.

We thank reviewer 2 for their comment. As suggested, we have revised the sentence in question to be more explicit about the inward shift and included some additional discussion. We also reword the second half the sentence as suggested.

2. Uncertainties always exist when using PSD evolution to infer the underlying physical mechanism, which naturally come from the always limited spatial/temporal resolution of measurements. The novel method of presenting PSD evolution in this study does help reduce such uncertainties, and to me the results do support the dominant role of local acceleration during this event. The authors did also discuss the potential ambiguities in the last paragraph of page 9. It is also worth noting that some high-profile papers, such as Reeves et al. (2013) Science paper, also used the growing PSD peak to infer the dominance of local acceleration though with such type of uncertainties. Overall, I do think that this study highlights the important role of local acceleration on ultra-relativistic electron energization, and demonstrates that a two-step process of local acceleration combining with inward radial diffusion, which is suggested by some previous studies, is not required in accelerating electrons to ultrarelativistic energies.

We thank reviewer 2 for their comment and glad to see that they agree with our conclusions.